

# On the performance of satellite-based observations of $CO_2$ in capturing the NOAA Carbon Tracker model and ground-based flask observations over Africa land mass

Anteneh Getachew Mengistu [1] and Gizaw Mengistu Tsidu[1,2]

[1]Addis Ababa University, Addis Ababa, Ethiopia
[2]Botswana International University of Science and Technology, Palapye, Botswana

**Correspondence:** Anteneh G (antenehgetachew7@gmail.com)

**Abstract.** Africa is one of the most data-scarce regions as satellite observation at the equator is limited by cloud cover and there are a very limited number of ground-based measurements. As a result, the use of simulations from models are mandatory to fill this data gap. A comparison of satellite observation with model and available in-situ observations will be useful to estimate the performance of satellites in the region. In this study, GOSAT $XCO_2$ is compared with the NOAA CT2016 and six flask

observations over Africa using five years of data covering the period from May 2009 to April 2014. Ditto for OCO-2 $XCO_2$ against NOAA CT16NRT17 and eight flask observations over Africa using two years of data covering the period from January 2015 to December 2016. The analysis shows that the $XCO_2$ from GOSAT is higher than $XCO_2$ simulated by CT2016 by 0.28 ppm whereas OCO-2 $XCO_2$ is lower than CT16NRT17 by 0.34 ppm on African landmass on average. The mean correlations of 0.83 and 0.60 and average RMSD of 2.30 and 2.57 ppm are found between the model and the respective datasets from

GOSAT and OCO-2 implying the existence of a reasonably good agreement between CT and the two satellites over Africa's land region. However, significant variations were observed in some regions. For example, OCO-2 $XCO_2$ are lower than that of CT16NRT17 by up to 3 ppm over some regions in North Africa (e.g., Egypt, Libya, and Mali ) whereas it exceeds CT16NRT17 $XCO_2$ by 2 ppm over Equatorial Africa (10 $^0$S - 10 $^0$N ). This regional difference is also noted in the comparison of model simulations and satellite observations with flask observations over the continent. For example, CT shows a better sensitivity

in capturing flask observations over sites located in Northern Africa. In contrast, satellite observations have better sensitivity in capturing flask observations in lower altitude island sites. CT2016 shows a high spatial mean of seasonal mean RMSD of 1.91 ppm during DJF with respect to GOSAT while CT16NRT17 shows 1.75 ppm during MAM with respect to OCO-2. On the other hand, low RMSD of 1.00 and 1.07 ppm during SON in the model $XCO_2$ with respect to GOSAT and OCO-2 are determined respectively indicating better agreement during autumn. The model simulation and satellite observations exhibit

similar seasonal cycles of $XCO_2$ with a small discrepancy over Southern Africa and during wet seasons over all regions.

## 1 Introduction

Changes in atmospheric temperature, hydrology, sea ice, and sea levels are attributed to climate forcing agents dominated by $CO_2$ (Santer et al., 2013; Stocker et al., 2013). However, understanding the climate response to anthropogenic forcing in a





more traceable manner is still difficult due to a major uncertainty in carbon-climate feedbacks (Friedlingstein et al., 2006). Part of this uncertainty is due to a lack of sufficient data on the regional and global carbon cycle. This is compounded with inappropriate modeling practices to capture spatiotemporal variability of the carbon cycle. These problems can be solved through strengthening carbon monitoring networks,setting up proper modelling and reducing uncertainties in satellite retrieval.

Models with appropriate physical and mathematical formulations and sufficiently constrained by observations, can be used to understand the spatio-temporal nature of atmospheric $CO_2$.

Towards this, a number of national and international efforts have been initiated in the recent past by different government and non-government agencies across the globe. Among these efforts, ground-based observations of greenhouse gas using Total Carbon Column Observing Network (TCCON) is a notable one since it provides accurate and high–frequency measurements

of column-integrated $CO_2$ mixing ratio. For example, it has been established that TCCON has a precision of 0.25% for measurements taken under clear sky conditions (Wunch et al., 2011). However, the number of TCCON sites is limited and can not establish an accurate $CO_2$ amount and flux on a subcontinental or regional scale. Moreover, some studies show that the large uncertainty is amplified due to the uneven global distribution of TCCON sites (Velazco et al., 2017). In addition, none of these ground-based observation networks were found in Africa land mass. However, there are few TCCON sites around the

continent plus some flask observations in and around Africa.

On the other hand, the $CO_2$ concentration retrieved from the satellite-based $CO_2$ absorption spectra have the advantages of being unified, long-term, and global observations as compared to ground-based measurements. It has been established from theoretical studies that accurate and precise satellite-derived atmospheric $CO_2$ can appreciably minimize the uncertainties in estimated $CO_2$ surface flux (Rayner and O'Brien, 2001; Chevallier, 2007). Other studies have revealed that significant

improvement in the estimation of weekly and monthly $CO_2$ fluxes can be achieved subject to $CO_2$ retrieval error of less than 4 ppm from satellite and modeling scheme whereby $CO_2$ concentration is an independent parameter of the carbon cycle model (Houweling et al., 2004; Hungershoefer et al., 2010). However, $XCO_2$ shows temporal variability on different time scales: diurnal, synoptic, seasonal, inter-annual, and long term (Olsen and Randerson, 2004; Keppel-Aleks et al., 2011). More recent missions such as the Greenhouse gases Observing SATellite (GOSAT) (Hamazaki et al., 2005), the Orbiting Carbon

Observatory-2 (OCO-2) (Boesch et al., 2011) and planned missions such as the Active Sensing of $CO_2$ Emissions over Nights, Days, and Seasons (ASCENDS) (Dobbs et al., 2008) have been and are being developed specifically to resolve surface sources and sinks of $CO_2$ and provide information on these different scales of temporal variability. For example, GOSAT observations started in 2009 and provide $XCO_2$ based on spectra in the Short-Wavelength InfraRed (SWIR) region with a standard deviation of about 2 ppm with respect to ground-based and in-situ air-borne observations (Yokota et al., 2009; NIES GOSAT Project,

2012). The bias and performance of $XCO_2$ retrievals from an algorithm could change in different regions with differing land surfaces and anthropogenic emissions.

Moreover, the NOAA Carbon Tracker (CT) is an integrated modeling system that assimilates $CO_2$ from other observations in order to complement satellite observations in understanding $CO_2$ surface sources and sinks as well as its spatiotemporal variabilities. However, both satellite and model data should be validated against other independent satellite observations and/or

in-situ observations before using them to answer scientific questions. As a result, a lot of validation and intercomparison have





been conducted in previous studies. For example, Kulawik et al. (2016) found root mean square deviation of 1.7, and 0.9 ppm in GOSAT and CT2013b $XCO_2$ relative to TCCON respectively. Other authors have undertaken validation exercises and found the bias of $-8.85 \pm 4.75\ ppm$ in retrieving $XCO_2$ from the GOSAT observed spectrum by Japans the National Institute for Environmental Studies (NIES) level 2 V02.xx $XCO_2$ (Yoshida et al., 2013) with respect to TCCON (Morino

et al., 2010). In addition, Chevallier (2015) shows retrieved $XCO_2$ from GOSAT observed spectrum by NASA Atmospheric $CO_2$ Observations from Space (ACOS) (O'Dell et al., 2012) suffers a systematic error over African Savanna. Lei et al. (2014) also showed a regional difference of $XCO_2$ between the ACOS and NIES datasets. For example, a larger regional difference from 0.6 to 5.6 ppm was obtained over China land region, while it is from 1.6 to 3.7 ppm over the global land region and from 1.4 to 2.7 ppm over US land region. These findings suggest that it is important to assess the accuracy and uncertainty

of $XCO_2$ from Satellite observations with respect to more accurate models (e.g., NOAA Carbon Tracker ) and ground-based observations over other regions as well. As satellite retrievals are strongly constrained by cloud cover, aerosol lodgings, land use change and Africa is a continent with wide extremes in surface type (which ranges from desert, rainforest and Savannah) and aerosol loading. Assessing the performance of satellites over the region can tell much about how these systematic errors vary geographically over the continent.

Therefore, this paper aims to assess the performance of observed $XCO_2$ from GOSAT and OCO-2 satellite in capturing simulated $XCO_2$ from NOAA Carbon Tracker model over Africa. These satellite observations and Carbon Tracker mixing ratios near the surface are also compared to available in suit $CO_2$ flask data from Assekrem, Algeria; Mt. Kenya; Gobabeb, Namibia; and Cape Town; as well as to data off the coast at Seychelles, Ascension Island, and at Izana, Tenerife. Moreover, the consistency between the model and satellite observations in capturing the amplitudes and phases of observed seasonal

cycles over different parts of the continent are evaluated. The agreement of modeled spatiotemporal variability with the known seasonal climatology of the regions, that determines carbon source and sink levels, is also assessed.

## 2    Data and Methodology

### 2.1    Carbon Tracker Model and Data

Carbon Tracker provides an analysis of atmospheric carbon dioxide distributions and their surface fluxes (Peters et al., 2007).

It is a data assimilation system that combines observed carbon dioxide concentrations from 81 sites around the world with model predictions of what concentrations would be based on a preliminary set of assumptions ("the first guess") about sources and sinks for carbon dioxide. Carbon Tracker compares the model predictions with reality and then systematically tweaks and evaluates the preliminary assumptions until it finds the combination that best matches the real world data. It has modules for atmospheric transport of carbon dioxide by weather systems, for photosynthesis and respiration, air-sea exchange, fossil fuel

combustion, and fires. Transport of atmospheric $CO_2$ is simulated by using the global two-way nested transport model (TM5). TM5 is an offline atmospheric tracer transport model (Krol et al., 2005) driven by meteorology from the European Centre for Medium-Range Weather Forecasts ($ECMWF$) operational forecast model and from the ERA-Interim reanalysis (Dee et al., 2011) to propagate surface emissions. TM5 is based on a global $3^0 \times 2^0$ and at a $1^0 \times 1^0$ spatial grids over North America.



CT date from the CT2015 release and on wards uses aircraft profiles from the stratosphere to the top of the atmosphere (Inoue et al., 2013; Frankenberg et al., 2016) and also co-location error are quantified (Kulawik et al., 2016). The older data versions have been used and also compared with different data sets over other parts of the globe in previous studies (Nayak et al., 2014; Kulawik et al., 2016). Most of the studies confirm that CT $XCO_2$ captures observations reasonably

well. In this study, we use Carbon Tracker release version CT2016, hereafter (CT2016) and near real-time version (CT-NRT.v2017). Both versions of NOAA CT provides 3 hourly $CO_2$ mole-fractions data for global atmosphere at 25 pressure levels in a $3^0 \times 2^0$ spatial resolution for a period covering 2000 to 2016. The data can be accessed freely at the public domain (ftp://aftp.cmdl.noaa.gov/products/carbontracker).

## 2.2 GOSAT measurements

GOSAT is the world's first spacecraft to measure the concentrations of carbon dioxide and methane, the two major greenhouse gases, from space. The spacecraft was launched successfully on January 23, 2009, and has been operating properly since then. GOSAT records reflected sunlight using three near-infrared band sensors. The field of view at nadir allows a circular footprint of about 10.5 km diameter (Kuze et al., 2009; Yokota et al., 2009; Crisp et al., 2012). GOSAT consists of two instruments. The sensors for the two instruments can be broadly labeled as thermal, near infrared and imager. The first two sensors are used as

part of Fourier Transform Spectrometer for carbon monitoring which is referred to as TANSO-FTS while the imager for cloud and aerosol observations is referred to as TANSO-CAI. The details on spectral coverage, resolution, field of view, and different products of TANSO-FTS in the three SWIR bands can be found in a number of previous studies (Kuze et al., 2009; Saitoh et al., 2009; Yokota et al., 2009, 2011; Crisp et al., 2012; Nayak et al., 2014; Deng et al., 2016a, and references therein). In this study ACOS B3.5 Lite $XCO_2$ from GOSAT Level 2 (L2) retrieval based on the SWIR spectra of FTS observations and

made available by Atmospheric $CO_2$ Observations from Space (ACOS) of NASA is used. ACOS B3.5 Lite $XCO_2$ has lower bias and better consistency than NIES GOSAT SWIR L2 $CO_2$ globally (Deng et al., 2016a). However, this version of ACOS $XCO_2$ found to suffer systematic retrieval error over the dark surfaces of high latitude lands and and over African savanna (Chevallier, 2015). Therefore, our choice of the ACOS B3.5 Lite, hereafter (GOSAT) $XCO_2$ is motivated by these differences.

## 2.3 OCO-2 measurements

OCO-2, the second world's full-time dedicated $CO_2$ measurement satellite. It was successfully launched by the National Aeronautics and Space Administration (NASA) on 2 July 2014. OCO-2 measures atmospheric carbon dioxide with the accuracy, resolution, and coverage required to detect $CO_2$ source and sink on global and regional scale. OCO-2 has three-band spectrometer, which measures reflected sunlight in three separate bands. The $O_2$ A-band measures molecular absorption of oxygen from reflected sunlight near 0.76 $\mu m$ while the $CO_2$ bands are located near 1.61 $\mu m$ and 2.06 $\mu m$ (Liang et al., 2017). In this

study, OCO-2 $XCO_2$ V7 lite level 2 covering the period from January 2015 to December 2016, hereafter referred to as OCO-2 $XCO_2$ are used. Due to the scarcity of data, CT values from the two releases CT2016 for the year 2015 and CT-NRT.v2017 for the year 2016, hereafter (CT16NRT17) are employed in this study. The OCO-2 project team at Jet Propulsion Laboratory,





California Institute of Technology, produced the OCO-2 $XCO_2$ data used in this study. The data can be accessed from NASA Goddard Earth Science Data and Information Service Center.

## 2.4 Flask observations

Measurements of $CO_2$ from nine ground-based flask observations near and within Africa land mass were accessed from the
NOAA/ESRL/GMD CCGG cooperative air sampling network https://www.esrl.noaa.gov/gmd/ccgg/flask.php. Sites description is given in Table 1.

**Table 1.** Information on flask observation sites near and within Africa land mass. * indicates discontinued site or project.

| Code | Name | country | Latitude ($^0N$) | Longitude ($^0E$) | Altitude (masl) | Air pressure at T = $25^0C$ (Pa) |
|------|------|---------|------------------|-------------------|-----------------|----------------------------------|
| ASC | Ascension Island | United Kingdom | -7.967 | -14.400 | 85.00 | 100342.02 |
| ASK | Assekrem | Algeria | 23.262 | 5.632 | 2710.00 | 73571.64 |
| CPT | Cape Point | South Africa | -34.352 | 18.489 | 230.00 | 98682.99 |
| IZO | Izana, Canary Islands | Spain | 28.309 | -16.499 | 2372.90 | 76650.84 |
| LMP | Lampedusa | Italy | 35.520 | 12.620 | 45.00 | 100803.63 |
| MKN* | Mt. Kenya | Kenya | -0.062 | 37.297 | 3644.00 | 65579.92 |
| NMB | Gobabeb | Namibia | -23.580 | 15.030 | 456.00 | 96141.54 |
| SEY | Mahe Island | Seychelles | -4.682 | 55.532 | 2.00 | 101301.78 |
| WIS | Weizmann, Ketura | Israel | 29.965 | 35.060 | 151.00 | 99584.09 |

## 2.5 Methods

The GOSAT and CT model $XCO_2$ time series used in this investigation span five years, ranging from May 2009 to April 2014. Atmospheric $CO_2$ concentrations of NOAA Carbon-Tracker have global coverage with a $3^0 \times 2^0$ Longitude/Latitude resolution
which covers 426 grid boxes in our study area. Satellite observations, however, is different from model assimilation, and have gaps because of various reasons (e.g., cloud and the observational mode of the satellite). As a result, there is no one to one spatiotemporal match between the two data sets. For example, $CO_2$ products from the two datasets are not directly comparable since CT is a 3 hourly smooth and regular grid dataset whereas GOSAT $XCO_2$ is irregularly distributed in space and time. Thus, the CT $CO_2$ is extracted on the time and location of GOSAT-$XCO_2$ data. Using the grid point of CT as a reference bin,
the corresponding GOSAT $XCO_2$ found within a rectangle of $1.5^0 \times 1.5^0$ with center at the reference bin and with a temporal mismatch of a maximum of 3 hrs is extracted. Moreover, CT has higher vertical resolutions than GOSAT. As a result, the two can not be directly compared. It is customary to smooth the high-resolution data (in this case CT) with averaging kernels and a priori profiles of the low-resolution satellite measurements (in this case GOSAT). In addition, due to a difference between CT and GOSAT on the number vertical levels, CT $CO_2$ is interpolated to vertical levels of GOSAT. The CT $XCO_2$ ($XCO_2^{model}$
) used in the comparison is computed from the interpolated CT $CO_2$ ($CO_2^{interp}$ ), pressure weighting function (w), $XCO_2$ a





priori ($XCO_{2a}$ ), column averaging kernel of the satellites retrievals (A) and a priori profile ($CO_{2a}$ ) of the retrievals as per procedure discussed by Rodgers and Connor (2003); Connor et al. (2008); O'Dell et al. (2012); Chevallier (2015); Jing et al. (2018) and given as:

$$XCO_2^{model} = XCO_{2a} + \sum_i w_i^T A_i * (CO_2^{interp} - CO_{2a})_i \tag{1}$$

where $i$ is the index of the satellite retrieval vertical level and $T$ is the matrix transpose. To compare the CT simulations and the Satellites observation with the flask observations, the vertical profile of the satellite and CT were extracted at the corresponding pressure level and location within a box of $1.5^0$.

Correlation coefficients (R), bias and root mean square deviation (RMSD) are used to assess the level of agreement between the two data sets. The mean bias determines the average deviations in $XCO_2$ between Carbon Tracker simulation and satellite

observations. In this work the bias at the $j^{th}$ grid point is computed as:

$$Bias_j = \frac{1}{n} \sum_{i=1}^{n} (S_i - O_i) \tag{2}$$

where $S_i$ and $O_i$ are CT and GOSAT $XCO_2$ values over the $j^{th}$ pixel at the $i^{th}$ time respectively. To quantify the extent to which $XCO_2$ of CT and GOSAT agree, the pattern correlations at the $j^{th}$ grid point are computed as:

$$R_j = \frac{\frac{1}{n} \sum_{i=1}^{n} (S_i - \bar{S})(O_i - \bar{O})}{\sqrt{\frac{1}{n} \sum_{i=1}^{n} (S_i - \bar{S})^2} \sqrt{\frac{1}{n} \sum_{i=1}^{n} (O_i - \bar{O})^2}} \tag{3}$$

where $\bar{S}$ and $\bar{O}$ are the mean values of $S_i$ and $O_i$ over the $j^{th}$ pixel. The root mean square deviation (RMSD) which shows the standard error of the model with respect the observation at the $j^{th}$ grid point is computed as :

$$RMSD_j = \sqrt{\frac{1}{n} \sum_{i=1}^{n} ((S_i - \bar{S}) - (O_i - \bar{O}))^2} \tag{4}$$

this is the centered pattern root mean squared (RMS) difference which is obtained from the RMS error after the difference in the mean has removed (Taylor, 2001).

Comparison with in situ flask observation is achieved in a way that the Carbon Tracker and satellite observations are taken at a corresponding pressure level of the in-situ flask observation (as mentioned in Table 1) in order to correspond to flux-towers surface observation. Further the datasets are re sampled to fit the flask observations in a $3^0 X 3^0$ window centered the flux-towers and to the available months were averaged.





## 3  Results and discussions

### 3.1  Comparison of $XCO_2$ mean climatology from NOAA CT2016 and GOSAT

The column-averaged mole fraction of $CO_2$ obtained from the NOAA Carbon Tracker model and GOSAT observation was compared. The results are based on 426 grid boxes uniformly distributed to cover the whole of Africa's land region. The

analysis was based on five years of daily data starting from May 2009 to April 2014.

Fig. 1 shows temporal average of CT2016 (Fig. 1a) and GOSAT (Fig. 1b) $XCO_2$ distribution. The major common spatial feature in the mean map of $XCO_2$ from GOSAT and CT2016 reanalysis is dipole structure characterized by high $XCO_2$ northward of equator and low $XCO_2$ southward of equator with the exception of Southern part of Congo (Fig. 1a) and southern part of Democratic republic of Congo (Fig. 1b) these are characterized by spatially anomalous high $XCO_2$. The

Southern Africa region is characterized by weak anthropogenic $CO_2$ emission and high $CO_2$ uptake by the vegetation. This contributed to the observed dipole distribution. Another important pattern is anomalous peak over the annual average location of the Inter-tropical convergence zone (ITCZ) (Fig. 1b) which appears to fade over Eastern Africa. This is in agreement with the fact that carbon stocks and net primary production per unit land area is higher over Equatorial Africa and decreases towards northward and southward of the equator over arid environments (Williams et al., 2007). However, Fig. 1b shows that GOSAT

observations has some limitations in simulating this spatial pattern in comparison to GOSAT.

Fig. 1c shows the mean difference (CT2016–GOSAT) $XCO_2$ which ranges from -4 to 2 ppm. The highest difference between the CT2016 and GOSAT $XCO_2$ (as high as -4 ppm) is observed over Northern part of Equatorial Africa (e.g., Guinea, Ghana, Nigeria, Central Africa, western Ethiopia and South Sudan, .etc.) which are also known for near-year-round rainfall and relatively dense vegetation. The regions are known for their rain forest. The likely explanation could be $XCO_2$ the mean (over

five years) climatology may be slightly positively biased due to fewer GOSAT observations as shown in Fig.1d. The strategy and methods for cloud screening in GOSAT retrievals could lead to a smaller number of observation in the equatorial region (Crisp et al., 2012; O'Dell et al., 2012; Yoshida et al., 2013; Chevallier, 2015; Deng et al., 2016b). The number of datasets used for comparison range from 14 to 4288 from the gridbox to gridbox with a spatial mean of 1109 data over the continent. Fig. 1c also shows CT2016 simulations are overall lower than the values of GOSAT observation over most regions with an exception

in Gabon, Congo, southern Kenya and southern Tanzania where CT2016 simulations are higher than GOSAT observation by more than 1 ppm. The spatial distribution of global atmospheric $CO_2$ is not uniform because of the irregularly distributed sources of $CO_2$ emissions, such as large power plant and forest fire, and biospheric assimilation as clearly noted above.

Fig. 2a shows differences between CT2016 and GOSAT $XCO_2$ ranges from -4 to 3 ppm. Out of 100% occurrence, more than 90% of observed differences are within ± 2 ppmv. The mean difference between CT2016 and GOSAT means is about

-0.27 ppm with the standard deviation of 0.98 ppm indicating better regional consistency and low potential outliers. Moreover, a negative mean of the difference implies that $XCO_2$ simulated from CT2016 is lower than that of GOSAT retrievals over Africa land mass.

Because of selection criteria which permits a difference of 3 degrees long and wide, the two datasets are not exactly at the same point. The impact of the relative distance between them should be assessed before performing any statistical comparison.




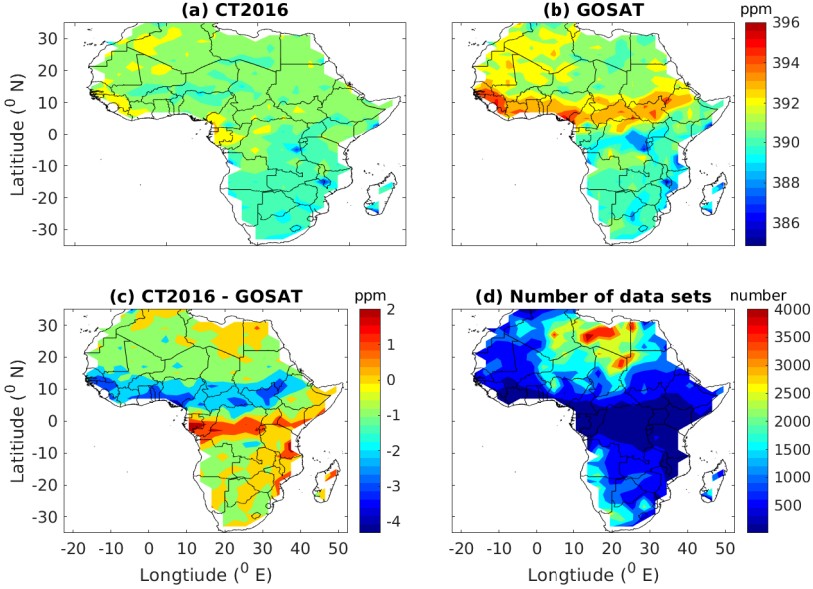

**Figure 1.** Distribution of five-years averages of CT2016 **(a)** and GOSAT **(b)** $XCO_2$ and their difference **(c)** gridded in $3^0 \times 2^0$ bins over Africa's Land mass; and the total number of datasets at each grid from the GOSAT observations(d).

Fig. 2b depicted color-coded scatter plot of CT2016 model simulation verses GOSAT to determine if the discrepancy between the datasets arise from the spatial mismatch. The color code indicates the relative distance between the model and observation datasets. For these datasets the $50^{th}$ percentile has a relative distance of $1.19^0$ which means 50% of the data has a relative distance of shorter than $1.19^0$. The maximum relative distance between them is $2.12^0$. However, there is no indication that this

has been the case since the scatter is not a function of the relative distance between the data sets. For example, data points with blue color with the lowest location difference is scattered everywhere instead of along the 1:1 line. Furthermore, we found the bias of -0.26 ppm, correlation coefficient of 0.86 and RMSD of 2.19 ppm for datasets which has a relative distance shorter than $1.19^0$. On the other hand, the bias, correlation coefficient, and RMSD are -0.33 ppm, 0.86 and 2.22 ppm for those which are above $1.19^0$. These statistics provide information there will be no strong discrepancy due to our selection criteria. The above

statistics was performed merely to test the influence of location mismatch.

  Fig. 3 shows a statistical comparison of $XCO_2$ from the CT2016 and GOSAT over Africa. The number of data used in this comparison is shown in Fig. 1d. As it is depicted in Fig. 3a, the bias ranges from -4 to 2 ppm with a mean bias of -0.28 ppm (see Table 2). A larger negative bias of about -2 ppm was found along with the annual mean position of ITCZ. The correlation varies from 0.4 over some isolated pockets in Congo, Tanzania, Mozambique, Uganda, and western Ethiopia to 0.9 over the northern

part of Africa above $13^0 N$, Eastern Ethiopia and the Kalahari Desert. Fig. 3b depicts correlation coefficient between GOSAT and Carbon Tracker $XCO_2$. The region with poor correlation also exhibits high RMSD as shown in Fig. 3c. To understand whether this discrepancy originates from model weakness alone, we have looked at the GOSAT posterior estimate of $XCO_2$





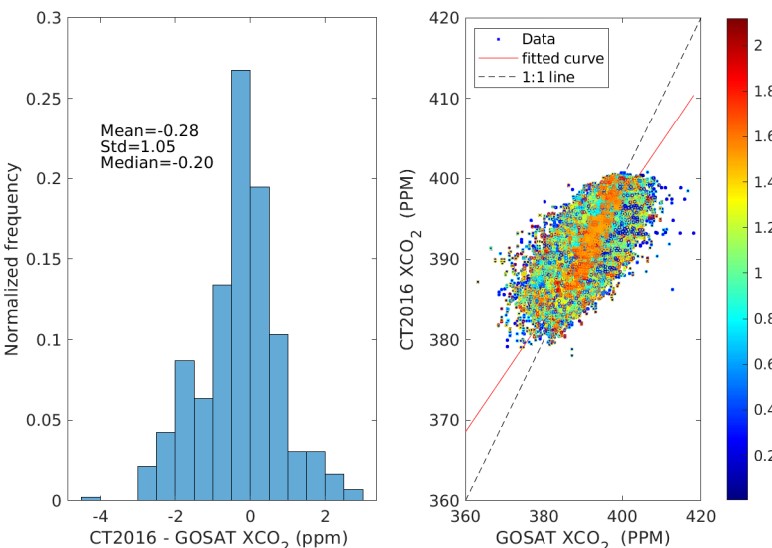

**Figure 2.** Histogram of the difference of CT2016 relative to GOSAT (left panel) and color code scatter diagram of $XCO_2$ concentration as derived from CT2016 and GOSAT (right panel). Color indicates the relative distance in unit of degrees as shown in colorbar between datasets.

error (Fig. 3d), which are high over regions where the bias and RMSD between GOSAT and Carbon Tracker $XCO_2$ is high. GOSAT's posterior estimate of $XCO_2$ error is a combination of instrument noise, smoothing error and interference errors (Connor et al., 2008; O'Dell et al., 2012). This posterior estimate of $XCO_2$ error does not include forward model error which may lead to underestimation of the true error of satellite $XCO_2$ by a factor of two (O'Dell et al., 2012). Therefore, part of
5  the discrepancy is clearly linked to satellite own uncertainty, which might have been amplified due to the small number of data points used to calculate the mean error of GOSAT $XCO_2$ measurements (see Fig. 1d). In general, the two data sets are characterized by a high spatial mean correlation of 0.83, a global offset of -0.28 ppm, which is the average bias, a regional precision of 2.30 ppm, which is average RMSD and relative accuracy of 1.05 ppm which is the standard deviation in the bias as depicted in Table 2.

**Table 2.** Summary of statistical relation between CT2016 and GOSAT observation. The statistical tools shown are the mean correlation coefficient (R), the spatial average of bias (Bias), the spatial average root mean square deviation (RMSD), the standard deviation in bias (std of Bias), GOSAT posteriori estimate of $XCO_2$ error (GOSAT err), the standard deviation in CT2016 $XCO_2$ (CT2016 std) and the standard deviation in GOSAT $XCO_2$ (GOSAT std). The number of data used in the statistics is 472,792 over 426 pixels covering the study period, distribution at each grid point is shown in Fig. 1d. Negative bias indicates that CT2016 $XCO_2$ is lower than GOSAT $XCO_2$ values.

| Statistical tool | R | Bias (ppm) | RMSD (ppm) | std of Bias (ppm) | GOSAT err (ppm) | CT2016 std (ppm) | GOSAT std(ppm) |
|---|---|---|---|---|---|---|---|
| Values | 0.83 | -0.28 | 2.30 | 1.05 | 0.91 | 0.90 | 1.55 |



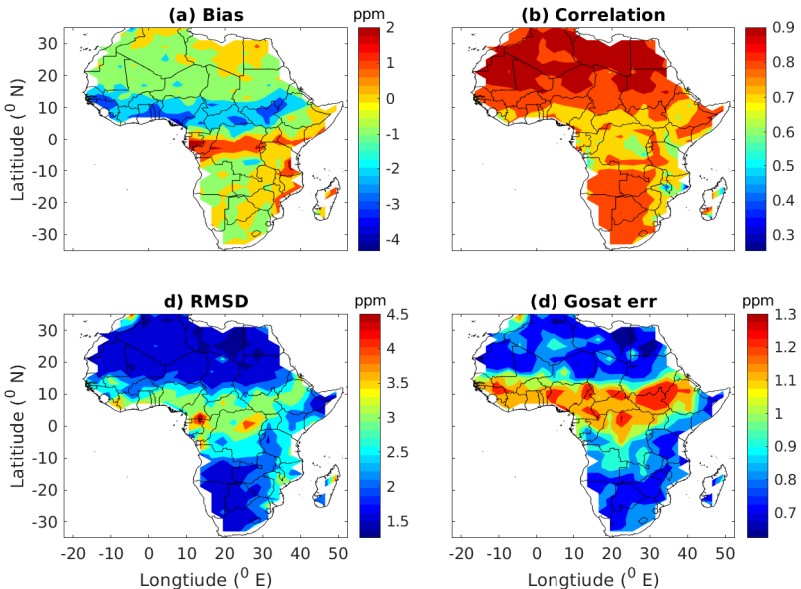

**Figure 3.** Spatial patterns of bias (a), correlation (b), RMSD (c) of the two data sets, and mean posteriori estimate of $XCO_2$ uncertainty from GOSAT (d).

## 3.2 Comparison of monthly average time series of NOAA CT2016 and GOSAT $XCO_2$

Africa is one of the largest continents covering both northern and southern hemispheres. As a result, the continent is under the influence of semi-permanent high-pressure cells which led to the Sahara Desert in the North and the Kalahari in the South. The equatorial low-pressure cell which allows the formation of the seasonally migrating inter-tropical convergence zone is part

of the major large scale atmospheric circulation systems. These large scale pressure systems, Oceanic circulations and their interaction with the atmosphere coupled with diverse topographies of the region allow for the formation of different climates (e.g., equatorial, tropical wet, tropical dry, monsoon, semi desert (semi arid), desert (hyper arid), subtropical high climates). Geographically, the Sahel, a narrow steppe, is located just south of Sahara; the central part of the content constitutes the largest rainforest next to Amazon whereas most southern areas contain savana plains. The continent gets rainfall from migrating

ITCZ, west Africa monsoon, the intrusion of mid-latitude frontal systems, travelling low pressure systems (Mitchell, 2001, and references therein). Since $CO_2$ fluxes exhibit seasonal variability and Africa experiences different seasons as noted above, it is important to divide Africa into three major regions, namely North Africa (10 to 35 $^0N$), Equatorial Africa (10 $^0S$ to 10 $^0N$), and Southern Africa (35 to 10 $^0S$) and conduct the comparison of the two $XCO_2$ datasets.

Figs. 4 - 6 show trends of monthly mean $XCO_2$ from CT2016 and GOSAT averaged over North Africa, Equatorial Africa,

and Southern Africa respectively. Figs. 4a - 6a depict the existence of an overall very good agreement for the monthly averages





**Table 3.** Summary of statistical relation between CT2016 and GOSAT observation. The statistical analysis was made using monthly averaged time series of 60 months (i.e., months from May 2009 to April 2014).

| Statistics | R | Bias (ppm) | RMSD (ppm) | number of data |
|---|---|---|---|---|
| Africa | 0.997 | -0.254 | 0.265 | 698505 |
| North Africa | 0.996 | -0.361 | 0.345 | 424070 |
| Equatorial Africa | 0.977 | -0.172 | 0.708 | 101660 |
| Southern Africa | 0.964 | 0.006 | 0.841 | 172775 |

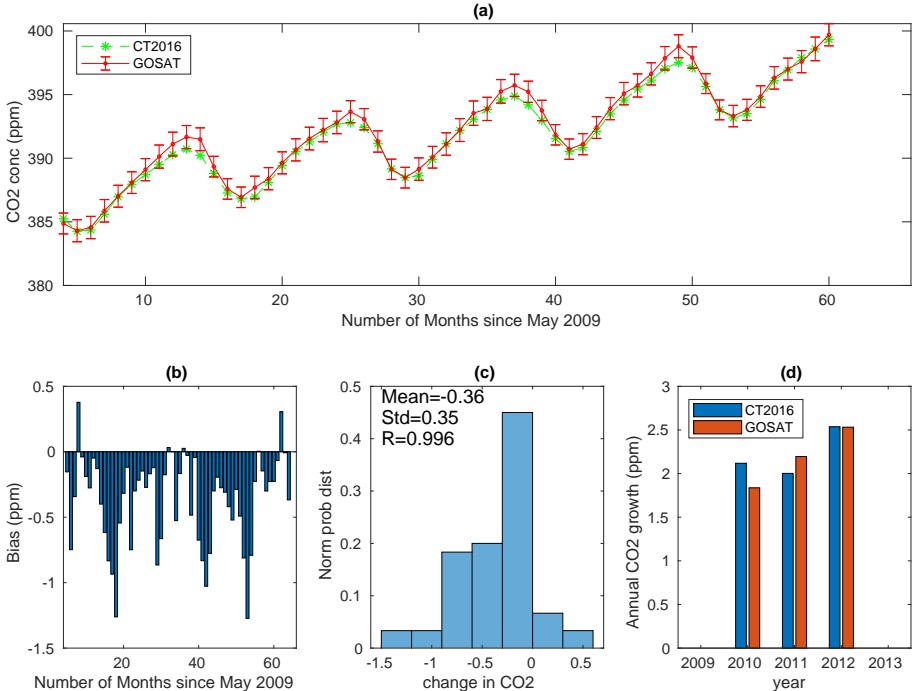

**Figure 4.** The monthly mean time series of CT2016 and GOSAT from May 2009 to April 2014 averaged over North Africa (a), bias associated with the monthly means (b), the histogram of difference (c) and the annual growth rate obtained by subtracting the mean from the mean of the next year (d). The error bars in (a) shows the GOSAT a posteriori $XCO_2$ uncertainty.

with respect to amplitudes and phase of $XCO_2$. However, $XCO_2$ from the two datasets slightly disagree in capturing seasonal cycle over Southern Africa.

Fig. 4a shows that $XCO_2$ concentration reaches maximum in April and minimum in September over North Africa. Consistent with this evidence, other authors (e.g., Zhou et al., 2008) have indicated the presence of strong absorption of $CO_2$ by vegetation during August in the northern hemisphere. This is the most likely cause for minimum concentration observed during September over North Africa. Both datasets show a concentration of $XCO_2$ increases from October to April and decreases



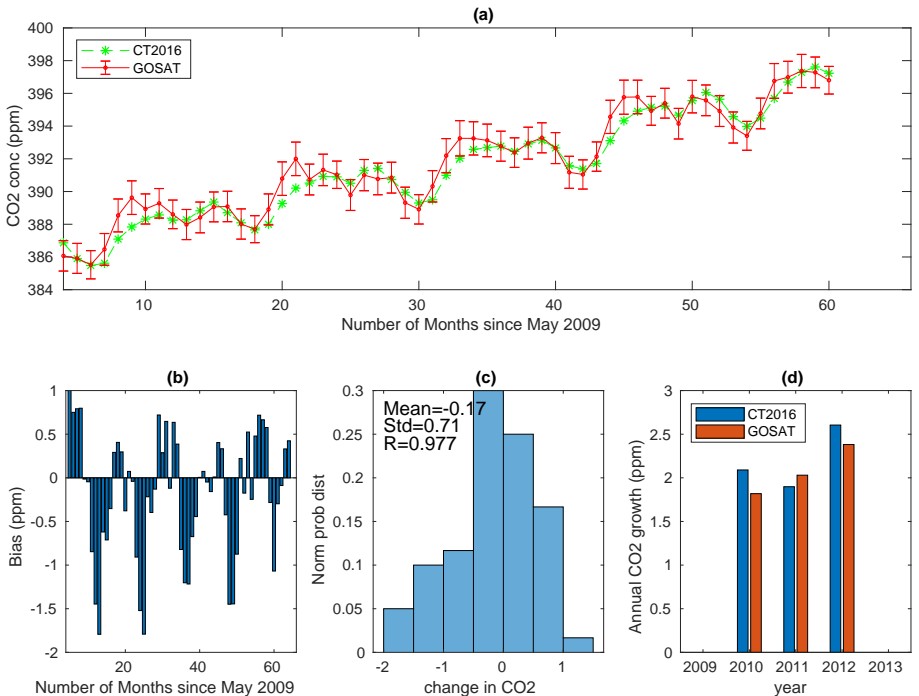

**Figure 5.** The same as Fig. 4 but over Equatorial Africa.

from May to September (see also Table 4). Moreover, the two dataset shows a monthly mean regional mean bias of -0.36 ppm with a correlation of 1.0 and small root mean square deviation of 0.36 ppm (see Table 3).

Fig. 5a shows $XCO_2$ concentration reaches maximum (392.99 ppm) for CT2016 in March and (393.53 ppm) for GOSAT in January while minimum (389.56 ppm for CT2016 and 389.32 ppm for GOSAT) in October over Equatorial Africa. The largest

monthly mean difference of -1.34 ppm and the smallest of -0.05 ppm between the two datasets observed in December and in April respectively (Table 4). Moreover, both datasets show that concentration of $CO_2$ increases from October to March while it decreases from June to October. This similarity in the seasonal variability of the two datasets shows that they are in good agreement in terms of amplitude and phase. In addition, the two datasets show a monthly average regional average bias of -0.17 ppm, correlation of 0.98 and a small root mean square deviation of 0.71 ppm over Equatorial Africa (see Table 3). Fig. 6a shows

maximum $XCO_2$ concentration in April (391.04 ppm) for CT2016 and in October (391.28 ppm) for GOSAT, while minimum in May (389.30 ppm) for CT2016 and ( 388.46 ppm) for GOSAT over Southern Africa. The largest monthly mean difference of 1.53 ppm and 0.03 ppm between the two datasets is observed in April and in July (Table 4) respectively. Both datasets show a concentration of $CO_2$ increases from May to July while it decreases from October and November. However, the $XCO_2$ from CT2016 shows a gradually increasing trend from January to April. Conversely, GOSAT $XCO_2$ shows decreasing values. This

is most likely CT2016 simulation respond to the growing size of sink following the rainy season. Moreover, the two datasets





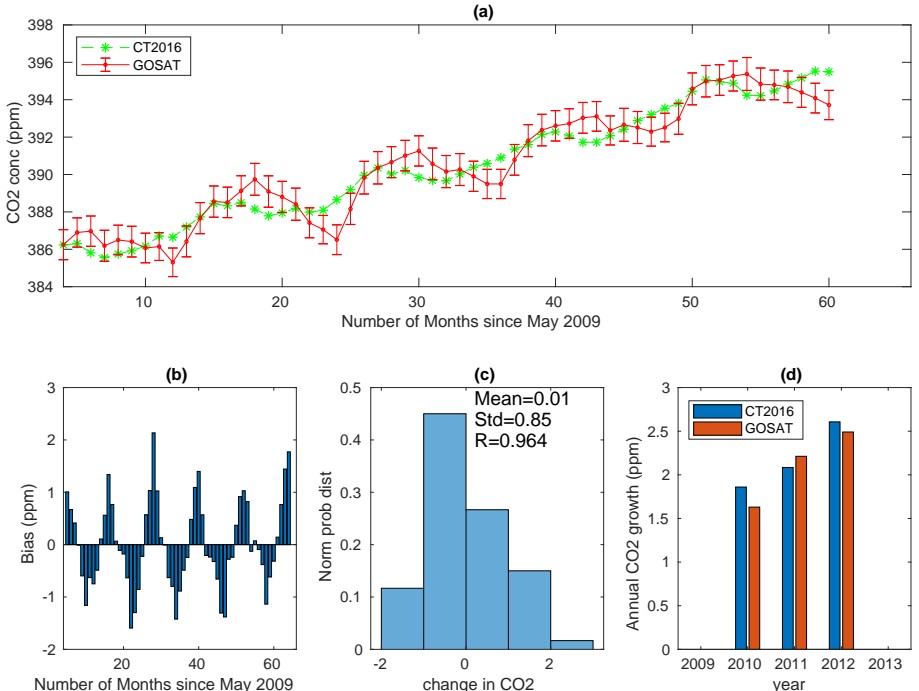

**Figure 6.** The same as Fig. 4 but over Southern Africa.

show a monthly mean regional mean bias of 0.07 ppm, correlation of 0.97 and RMSD of 0.87 ppm over southern Africa (see Table 3).

Figs. 4b - 6b show regional averaged bias in the monthly mean $XCO_2$ from CT2016 and GOSAT. Fig. 4b shows the presence of seasonally varying negative bias over North Africa. A high (<-0.5 ppm) negative bias in dry seasons (April to June) and

low (>=-0.1 ppm) negative bias in wet seasons (August to September) are observed. Moreover, the strength of bias increases from February to June. Conversely, the bias decreases from June to September. Similarly, Figs. 5b and 6b show seasonally fluctuating bias. For example, Fig. 6b shows a positive bias from February to July and negative bias from August to December over Southern Africa.

Figs. 4c - 6c show the histogram of difference. The mean difference between CT2016 simulation and GOSAT observation of

$XCO_2$ is -0.36 ppm with a standard deviation of 0.35 ppm over North Africa (see Fig. 4c); Fig. 5c presents a mean difference of -0.17 ppm with a standard deviation of 0.71 ppm over Equatorial Africa and Fig. 6c reveals a mean difference of 0.01 ppm and a standard deviation of 0.85 ppm which indicates that $XCO_2$ from CT2016 was slightly higher than that of GOSAT over Southern Africa on average. In addition, the low standard deviation of monthly mean difference over North Africa typically indicates good regional consistency between CT2016 and GOSAT. This is mainly because Northern Africa is dominated by

the Sahara desert which is known for its weak source/sink of $CO_2$. However, the spatial mean of monthly mean bias is slightly higher (-0.36 ppm) over North Africa than over Equatorial Africa (-0.17 ppm ) and Southern Africa (0.01 ppm). This is likely





**Table 4.** Five years monthly averaged $XCO_2$ concentration in ppm obtained from CT2016 (CT) and GOSAT (GO) and their difference $CT - GO$ (D) in ppm over Africa (A), North Africa (NA), Equatorial Africa(EA) and Southern Africa (SA).

| Month | A CT | A GO | A D | NA CT | NA GO | NA D | EA CT | EA GO | EA D | SA CT | SA GO | SA D |
|---|---|---|---|---|---|---|---|---|---|---|---|---|
| January | 391.81 | 392.17 | -0.36 | 392.43 | 392.61 | -0.18 | 392.22 | 393.53 | -1.31 | 390.28 | 390.49 | -0.21 |
| February | 392.48 | 392.58 | -0.1 | 393.27 | 393.5 | -0.23 | 392.72 | 393.21 | -0.49 | 390.52 | 390.06 | 0.46 |
| March | 393.25 | 393.28 | -0.03 | 394.02 | 394.29 | -0.27 | 392.99 | 393.19 | -0.2 | 390.82 | 389.81 | 1.01 |
| April | 393.81 | 393.91 | -0.1 | 394.79 | 395.35 | -0.56 | 392.87 | 392.92 | -0.05 | 391.04 | 389.51 | 1.53 |
| May | 391.65 | 391.85 | -0.21 | 392.92 | 393.73 | -0.81 | 390.47 | 389.93 | 0.54 | 389.3 | 388.46 | 0.84 |
| June | 391.49 | 391.94 | -0.45 | 392.43 | 393.33 | -0.9 | 391.12 | 390.89 | 0.23 | 389.95 | 389.85 | 0.11 |
| July | 390.92 | 391.1 | -0.18 | 391.09 | 391.5 | -0.41 | 391.44 | 391.03 | 0.41 | 390.43 | 390.4 | 0.03 |
| August | 389.89 | 389.96 | -0.07 | 389.4 | 389.44 | -0.04 | 390.92 | 390.72 | 0.21 | 390.37 | 390.61 | -0.25 |
| September | 389.26 | 389.4 | -0.14 | 388.65 | 388.75 | -0.1 | 390.02 | 389.67 | 0.35 | 390.39 | 391.01 | -0.61 |
| October | 389.19 | 389.71 | -0.51 | 388.85 | 389.26 | -0.41 | 389.56 | 389.32 | 0.24 | 389.95 | 391.28 | -1.32 |
| November | 389.97 | 390.43 | -0.46 | 390.06 | 390.32 | -0.26 | 389.86 | 390.52 | -0.66 | 389.8 | 390.76 | -0.96 |
| December | 391.09 | 391.53 | -0.45 | 391.42 | 391.6 | -0.18 | 391.23 | 392.57 | -1.34 | 389.98 | 390.52 | -0.54 |

due to the presence of strong local source from emissions and long-range transport from the Northern Hemisphere as reported in other studies (Williams et al., 2007; Carré et al., 2010).

Figs. 4d - 6d display annual growth rate of $XCO_2$ which ranges from 1.5 to 2.7 $ppm\ yr^{-1}$. Moreover, the two datasets are consistent in determining the annual growth rate. The results are found in good agreement with the observed variability in the global annual growth rate from surface measurements (http://www.esrl.noaa.gov/ gmd/ccgg/trends/global.html) which is 1.67, 2.39, 1.70, 2.40, 2.51 ppm $yr^{-1}$ global during 2009 - 2013 respectively, and 1.89, 2.42, 1.86,2.63, 2.06 ppm $yr^{-1}$ for Mauna Loa during 2009 - 2013 respectively, with error bars of 0.05 - 0.09 ppm $yr^{-1}$ for global and 0.11 ppm $yr^{-1}$ for Mauna Loa data sets(Kulawik et al., 2015). The growth rate may not be conclusive due to the short length of the datasets used. However, it reflects how the CT and GOSAT observations perform with respect to each other.

## 3.3 Comparison of seasonal climatology

The seasonal cycle has important implications for flux estimates (Keppel-Aleks et al., 2012). It is important to analyze whether there are seasonally dependent biases that are affecting the seasonal cycle and whether the data sets are capturing the same seasonal cycle. The four seasons considered here are winter (December, January and February or in short DJF), spring (March, April and May or in short MAM ), summer (June, July and August or in short JJA), and autumn (September, October and November or in short SON). Fig. 7 shows the seasonal distributions of CT2016 (left panels) and GOSAT (middle panels) $XCO_2$ and their difference (CT2016 - GOSAT, right panels). The distribution clearly shows that $XCO_2$ concentration is maximum during spring (MAM) and minimum during autumn (SON) over the North Africa. On the other hand, maxima is found during autumn (SON) and minima during winter (DJF) over the Southern Africa. These features are in good agreement with





the rainfall climatology of northern and southern hemispheres. Moreover, Table 5 shows seasonally varying biases. Seasonal biases affect the seasonal cycle and amplitudes, which are important for biospheric flux attribution (Lindqvist et al., 2015).

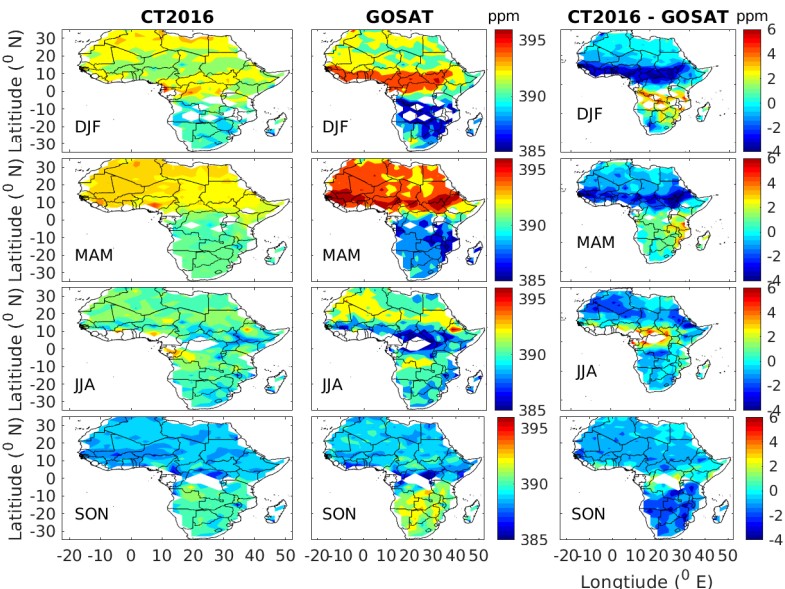

**Figure 7.** Seasonal climatology of $XCO_2$ for NOAA CT2016 (left panels) and GOSAT (midel panels) and their difference (right panels).

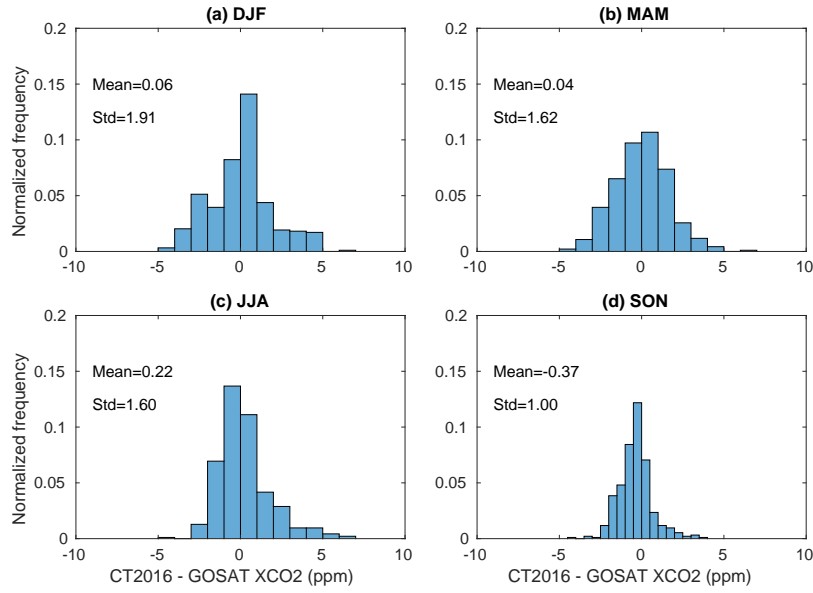

**Figure 8.** Histogram of difference for the seasonal $XCO_2$ climatology for DJF (a), MAM(b), JJA (c) and SON (d) seasons.





**Table 5.** Summary of statistical relation between CT2016 and GOSAT $XCO_2$: Bias, correlation (R), Root mean square deviation (RMSD), standard deviation of $XCO_2$ from CT2016 simulation (CT2016 std), standard deviation of $XCO_2$ from GOSAT observation (GOSAT std), aggregate number of coincident observations (number of data) and number of grids over the region (grid). Negative bias means CT2016 is lower than GOSAT. The statistics are on the basis of spatial average of seasonal averages of bias, correlation, RMSD and standard deviations.

| Region | Statistics | Bias (ppm) | R | RMSD (ppm) | CT2016 std (ppm) | std in GOSAT (ppm) | number of data | grid |
|---|---|---|---|---|---|---|---|---|
| Africa | DJF | 0.06 | 0.73 | 1.91 | 1.15 | 2.57 | 135865 | 409 |
| | MAM | 0.04 | 0.92 | 1.62 | 1.98 | 3.25 | 95942 | 410 |
| | JJA | 0.22 | 0.65 | 1.59 | 1.12 | 2.08 | 116360 | 400 |
| | SON | -0.37 | 0.76 | 1 | 0.94 | 1.52 | 124233 | 408 |
| North Africa | DJF | -0.25 | 0.36 | 1.08 | 0.67 | 1.12 | 103913 | 204 |
| | MAM | -0.72 | 0.44 | 1.11 | 0.62 | 1.24 | 65115 | 204 |
| | JJA | -0.42 | 0.73 | 1.17 | 0.9 | 1.66 | 60854 | 204 |
| | SON | -0.35 | 0.66 | 0.53 | 0.52 | 0.71 | 91778 | 204 |
| Equatorial Africa | DJF | -0.52 | 0.68 | 2.47 | 1.06 | 3.07 | 22639 | 121 |
| | MAM | 0.18 | 0.9 | 1.88 | 1.94 | 3.46 | 8300 | 115 |
| | JJA | 1.51 | 0.59 | 2.02 | 1.46 | 2.52 | 12714 | 104 |
| | SON | 0.25 | 0.7 | 1.3 | 1.16 | 1.83 | 10213 | 113 |
| Southern Africa | DJF | 1.61 | 0.42 | 1.72 | 0.88 | 1.9 | 9313 | 84 |
| | MAM | 1.56 | 0.67 | 0.97 | 0.82 | 1.31 | 22527 | 91 |
| | JJA | 0.18 | 0.81 | 0.78 | 0.93 | 1.31 | 42792 | 92 |
| | SON | -1.16 | 0.77 | 0.81 | 0.84 | 1.26 | 22242 | 91 |

The right panels in Fig. 7 show that the seasonal mean difference (CT2016 - GOSAT) ranges from -4 to 6 ppm. A maximum difference of 6 ppm over the Gulf of Guinea and Congo during JJA. However, such maximum difference was also observed over Southern Africa during DJF. A minimum of -4 ppm over annual mean ITCZ region was observed during DJF and MAM. Moreover, the difference is above 1 ppm over Southern Africa regions during DJF and MAM (wet season of the region). This implies high spatial variability of the seasonal mean difference during different seasons (see also Table 5). It also suggests that the discrepancy between the CT2016 and GOSAT becomes significant when vegetation cover is weak during DJF and MAM (dry seasons) over North Africa.

During SON the seasonal difference in most Africa's land region ranges from -2 to 1 ppm. The result implies CT2016 simulates lower values of $XCO_2$ than that of GOSAT observation indicating that there is a better spatial consistency during this season. Furthermore, during these seasons both the Northern and Southern Africa have a moderate vegetation cover following





their respective summer seasons. The two datasets show lower regional variation (i.e., only from -2 to 2 ppm) over most of Africa land mass. However, Equatorial Africa exhibits the mean difference lower than -2 ppm during DJF and MAM. This indicates the model tends to simulate lower than GOSAT retrievals $XCO_2$ over the region. In addition, this strong negative bias is partially due to a positive bias in GOSAT $XCO_2$ retrieval due to cirrus clouds. For example,O'Dell et al. (2012) noted

that GOSAT $XCO_2$ retrievals are positively biased due to thin cirrus clouds. Fig. 7(right panels) reveals $XCO_2$ from CT2016 is lower than GOSAT $XCO_2$ over Northern Africa. The underestimation of observed $XCO_2$ by NOAA CT2016 model is likely related to the skill of driving ERA-Interim data as noted from previous studies. For example, Mengistu Tsidu (2012) has shown that the ERA-Interim data has a wet bias over Ethiopian highlands. Mengistu Tsidu et al. (2015) have also shown that ERA-Interim precipitable water is higher than measurements from radio-sonde, FTIR and GPS observations. Therefore,

such wet bias in the driving ERA-Interim GCM might have forced NOAA CT2016 to generate dense vegetation which serves as $CO_2$ sink. In another study, Nagarajan and Aiyyer (2004) found ECMWF has a cold bias in the lower atmosphere between 1000 to 750 hPa against independent upper-air sounding data which may affect $CO_2$.

Fig. 8 shows the mean difference between CT2016 and GOSAT $XCO_2$ seasonal means which ranges from -0.37 to 0.04 ppm with a standard deviation within a range of 1.00 to 1.91 ppm over the continent. The highest mean difference of $XCO_2$

(-0.37 ppm) occurs during SON and the lowest (0.04 ppm) occurs during MAM. Table 5 presents the summary of statistical values for the spatial mean of each season means. The comparison between the two data sets also shows there is a strong correlation (>0.5) during each season over the continent. However, there are moderate correlations (0.3 to 0.5) during DJF and MAM over North Africa and during DJF over Southern Africa. The low correlation over Northern Africa may be linked to a weak absorption by vegetation and a strong emission from human activities during winter as reported elsewhere (Liu et al.,

2009; Kong et al., 2010). Moreover, Table 5 shows that the seasonal biases are negative over North Africa while they are mostly positive over Equatorial and Southern Africa. Negative biases are observed during DJF and SON over Equatorial and Southern Africa respectively implying that $XCO_2$ from CT2016 are lower than GOSAT during dry seasons.

### 3.4 Comparison of GOSAT and CT2016 with flask observations

Comparison of GOSAT and CT2016 with flask observation are carried out over six available ground-based flask observations.

For the comparison, the volume mixing ratio of $CO_2$ from GOSAT and CT2016 at the pressure level that corresponds to surface observation of flask (see Table 1 ) were considered.

Monthly mean $CO_2$ from flask observations at IZO and ASK in northern Africa shows an excellent agreement with both CT2016 and GOSAT $CO_2$. Moreover, CT2016 has a better sensitivity in capturing the amplitudes than GOSAT where observations from GOSAT mostly under estimates higher values of flask $CO_2$ (Fig. 9). However, this agreement has deteriorated over

sites in Equatorial Africa (ASC and MKN) and Southern Africa (MNB). Over MKN, CT2016 shows better correlation (0.43) than GOSAT observation (0.08). In addition, monthly amplitudes from CT2016 was closer to the flask observations suggesting that satellite retrievals need much attention over the region. On the other hand, GOSAT observations were found to be in better agreement with flask observations over ASC. Zhang et al. (2015) also show that GOSAT data was correlated well with ground observation and found to be more centralized, having high system stability, especially over the ocean.





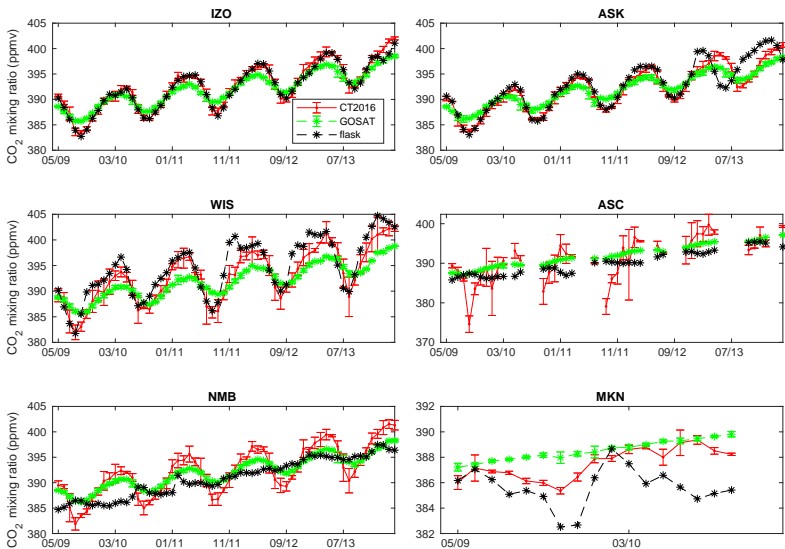

**Figure 9.** $CO_2$ time series for the coincident period for CT2016 (red), GOSAT (green) and flask (black). The standard deviation in computing the monthly mean is indicated by the vertical error bar.

**Table 6.** Summary of statistical relations of CT2016 and GOSAT observation with respect to flask observations. The statistical analysis was made using monthly averaged covering the period from May 2009 to April 2014).

| code | CT R | GOSAT R | CT Bias (ppm) | GOSAT Bias (ppm) | CT RMSD (ppm) | GOSAT RMSD (ppm) | number of data |
|------|------|---------|---------------|------------------|---------------|------------------|----------------|
| ASC | 0.58 | 0.93 | 1.05 | 1.84 | 4.46 | 1.07 | 39 |
| ASK | 0.90 | 0.90 | -0.63 | -0.76 | 1.97 | 2.23 | 60 |
| NMB | 0.75 | 0.91 | 1.40 | 1.13 | 3.12 | 1.56 | 60 |
| IZO | 0.99 | 0.97 | 0.24 | -0.36 | 0.70 | 1.40 | 60 |
| MKN | 0.40 | 0.04 | 1.83 | 2.88 | 1.48 | 1.64 | 17 |
| WIS | 0.93 | 0.83 | -1.57 | -2.61 | 1.95 | 3.31 | 60 |

CT2016 has a better sensitivity over IZO, ASK and NMB. Moreover, CT2016 compared well with flask observations than GOSAT over these sites, almost all flask observations are within the standard deviations of the monthly mean of CT2016. However, GOSAT observations were found in better agreement with flask observations than CT2016 was over WIS and ASC. On the other hand, both CT2016 and GOSAT have low sensitivity to flask observation over MKN (see Fig. 10). Similar to our previous discussion over sites in the Northern Africa (IZO, ASK and WIS), CT2016 underestimates $XCO_2$ during August, September, and October (wet season) compered to GOSAT observation and overestimates during January to June. However, the CT2016 and the flask observations exhibit better agreement indicating a bias in GOSAT observation during the wet season.





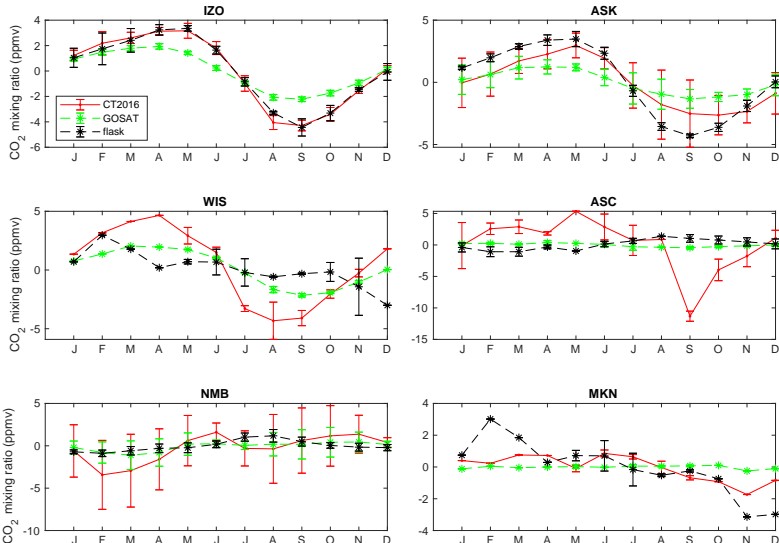

**Figure 10.** De-trended seasonal cycle of $XCO_2$ during 2009-2014 from CT2016 (red), GOSAT (green) and flask (black) observations. The standard deviation of the monthly variables is indicated by error bars.

### 3.5 Comparison of mean $XCO_2$ from NOAA CT16NRT17 and OCO-2

The strong El Niño event occurred during 2015-2016 provides an opportunity to compare the performance of CT16NRT17 during strong El Niño events. Because of the decline in terrestrial productivity and enhancement of soil respiration, the concentration of $CO_2$ increases during El Niño events (Jones et al., 2001). In this section we compare mean $XCO_2$ of NOAA

CT16NRT17 and NASA's OCO-2 covering the period from January 2015 to December 2016.

The comparison was done based on the selection criteria discussed in Section 2.5. Fig. 11 shows mean distribution of $XCO_2$ from CT16NRT17 (Fig. 11a) and OCO-2 (Fig. 11b) over Africa's land mass. CT16NRT17 shows high ( $> 400$ ppm) $XCO_2$ values over North Africa while these high $XCO_2$ values are observed over Equatorial Africa in the case of OCO-2 observation. The two datasets show a discrepancy over Equatorial Africa, where CT16NRT17 simulates low $XCO_2$ values (< 401 ppm)

while OCO-2 observes high values of $XCO_2$ (> 401 ppm). Both datasets show moderate $XCO_2$ values which ranges from 397 to 400 ppm over Southern Africa. The $XCO_2$ distribution from OCO-2 is consistent with the maximum $CO_2$ concentration reported in past study by Williams et al. (2007) implying that the CT16NRT17 likely underestimates $XCO_2$ values over Equatorial Africa. It is also possible that the discrepancy is a compounded effect of OCO-2 $XCO_2$ positive bias over the region (O'Dell et al., 2012; Chevallier, 2015). Fig. 11c shows the mean difference between two years mean of $XCO_2$ from

CT16NRT17 and OCO-2, which is in the range from -2 to 2 ppm. However, high (<-2 ppm) negative mean difference between the two data sets over rain forest regions (Gulf of Guinea and Congo basin) and ITCZ zone over Eastern Africa (South Sudan and southeastern Sudan) is observed implying that CT16NRT17 simulates lower $XCO_2$ values than that of OCO-2 observation over regions where vegetation uptake is strong. Conversely, high (>1) positive mean difference over the Sahara desert, Somalia





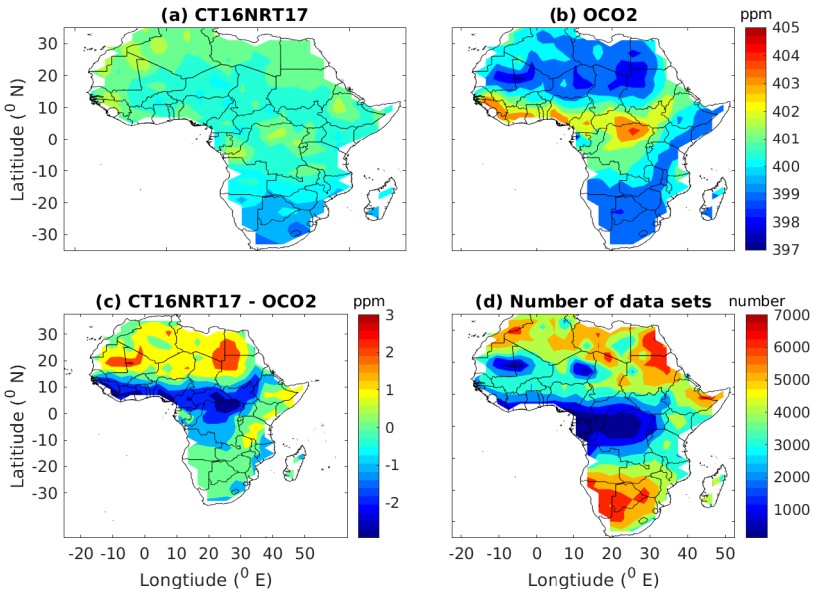

**Figure 11.** Distribution of two years average $XCO_2$ of CT16NRT17 **(a)** and OCO-2 **(b)** $XCO_2$ and their difference **(c)** gridded in $3^0 \times 2^0$ bins; and **(d)** the total number of datasets at each grid

and Tanzania implies CT16NRT17 simulates higher $XCO_2$ values than OCO-2 observation where the vegetation uptake is weak. Moreover, a positive (>2) mean difference over Egypt, Libya, Sudan, Chad, Niger, Mali and Mauritania is likely due to overestimates of $XCO_2$ emission from local sources by CT16NRT17. Overall, the two datasets show a fairly reasonable agreement with a correlation of 0.60 and offset of 0.36 ppm, a regional precision of 2.51 ppm and a regional accuracy of 1.21 ppm.

**Table 7.** Summary of statistical relation between CT16NRT17 and OCO-2 observation. The statistical tools shown are the mean correlation coefficient (R), the average of bias (Bias), the average root mean square deviation (RMSD), the standard deviation in bias (std of Bias), mean posteriori estimate of $XCO_2$ error from OCO-2 (OCO-2 err), the standard deviation in CT16NRT17 $XCO_2$ (CT16NRT17 std) and the standard deviation in OCO-2 $XCO_2$ (OCO-2 std). Positive Bias indicates that CT16NRT17 is higher than OCO-2. The number of data used in the statistics is 1,659,411 over 426 pixels covering the study period, distribution at each grid point is shown in Fig 11d.

| Statistical tool | R | Bias (ppm) | RMSD (ppm) | std of Bias (ppm) | OCO-2 err (ppm) | CT16NRT17 std (ppm) | OCO-2 std (ppm) |
|---|---|---|---|---|---|---|---|
| Values | 0.6 | 0.34 | 2.57 | 1.21 | 0.55 | 0.55 | 1.28 |

Fig. 12a shows the histogram of two years mean difference, which is characterized by a positive mean of 0.34 ppm and a standard deviation of 1.21 ppm. This suggests that CT16NRT17 simulates high $XCO_2$ as compared to observations from OCO-2 over Africa's land mass.





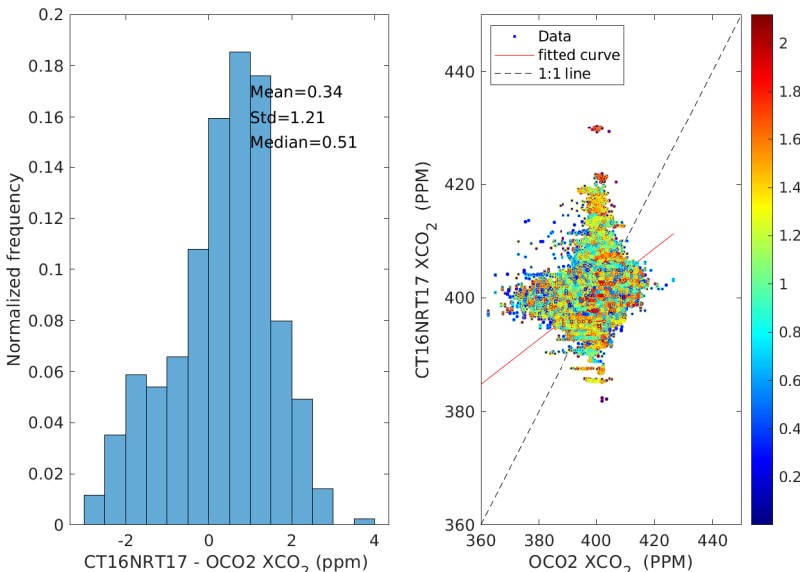

**Figure 12.** Histogram of the difference of CT16NRT17 relative to OCO-2 (left panel) and color code scatter diagram of $XCO_2$ concentration as derived from CT16NRT17 and OCO-2 (right panel). Color indicates the relative distance in unit of degrees as shown in colorbar between datasets.

Because of presence of spatial and temporal mismatch of some level between CT16NRT17 and OCO-2 datasets, it is important to assess the effect of relative distance between the datasets. Fig. 12b shows a color coded distribution of the two datasets. In the figure color codes indicate the relative distance. The random scatter of blue dots implies that the statistical discrepancies do not arise from the relative distance between the two datasets. More specifically, a statistical comparison of datasets lower

and higher the $50^{th}$ percentile ($1.2^0$) shows bias of 0.58 and 0.57 ppm, correlation of 0.57 and 0.57 and RMSD of 2.65 and 2.67 ppm respectively.

Fig. 13 shows the comparison of mean $XCO_2$ from CT16NRT17 and OCO-2 covering the period from January 2015 to December 2016. The number of data used are displayed in Fig. 11d. Fig. 13a depicts the bias which ranges from -2 to 2 ppm with a mean bias of 0.34 ppm. However higher biases (<-2 ppm) are observed over Equatorial Africa along the annual average

location of ITCZ. Fig. 13b shows the correlation map with values from 0.2 to 0.8 over Africa's land mass. A good correlation of above 0.6 are seen over many regions of the continent while weak correlation of less than 0.2 and higher root mean square error (> 3 ppm ) are observed over small pockets of Equatorial and Eastern Africa regions (see Fig. 13c). These regions also show a higher (> 0.65 ppm) error in satellite retrieval (see Fig. 13d). In addition, Fig. 11d shows the number of observations are small (< 1000 ) over these regions. This may contribute to the observed discrepancy over these regions. However, weak

correlations are also observed over a wider area in North Africa such as Mauritania, Mali, Algeria and some regions of Niger





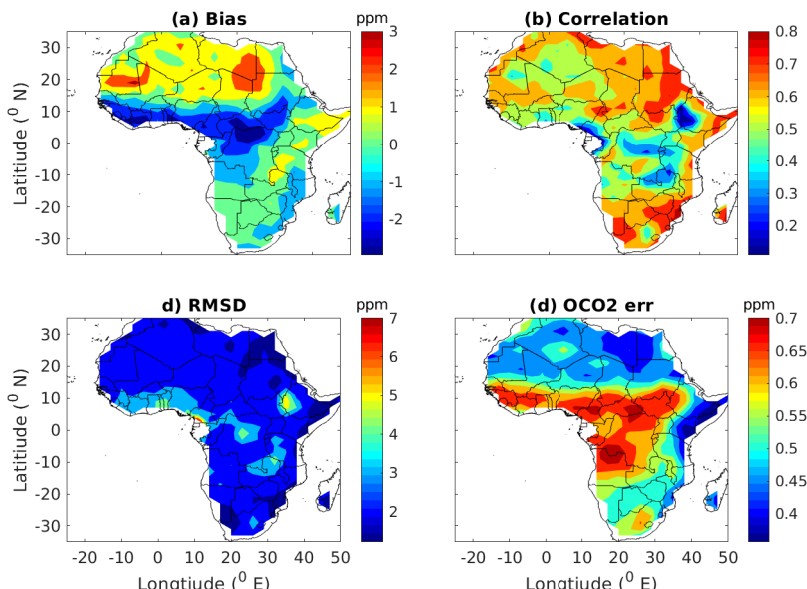

**Figure 13.** The bias (a), correlation (b), RMSD (c) of model and OCO-2 $XCO_2$ and mean posteriori estimate of $XCO_2$ error from OCO-2 (d).

where satellite errors are low and sufficient data are obtained. Poor correlation and higher RMSD values are observed over Southwest Ethiopia.

### 3.6 Comparison of monthly average time series of NOAA CT16NRT17 and OCO-2 $XCO_2$

Figs. 14 - 16 show a two year monthly average time series comparison of $XCO_2$ from CT16NRT17 and OCO-2 over North
5 Africa, Equatorial Africa and Southern Africa respectively. Fig. 14a shows the existence of good agreement between the two datasets in describing pattern over North Africa. Moreover, both datasets show a decreasing trend of $XCO_2$ from May to September while increasing trend from October to April. On the other hand, consistent with the climate condition and associated $CO_2$ exchange, the monthly mean $XCO_2$ shows a maximum value of 403.37 ppm for CT16NRT17 and 402.06 ppm for OCO-2 during May. Conversely, a minimum concentration of 398.77 ppm from CT16NRT17 simulation and 398.27
10 ppm from OCO-2 observation are found in September. In addition, both CT16NRT17 and OCO-2 show maximum $XCO_2$ values (402.15 ppm for CT16NRT17 and 402.03 ppm for OCO-2) in December. These pick values in December are not surprising, because the 2015-2016 El Niño started on March 2015 and reached pick in December 2015 which added extra $CO_2$ into the atmosphere (Chatterjee et al., 2017). Fig. 14a also shows that $XCO_2$ from CT16NRT17 simulation are higher than OCO-2 observation over North Africa.

15 Fig. 14b shows the monthly mean difference between CT16NRT17 and OCO-2 which ranges from -0.5 to 2 ppm. OCO-2 $XCO_2$ observations are lower than CT16NRT17 by 2 ppm during March and April 2015. Starting from August 2015, the



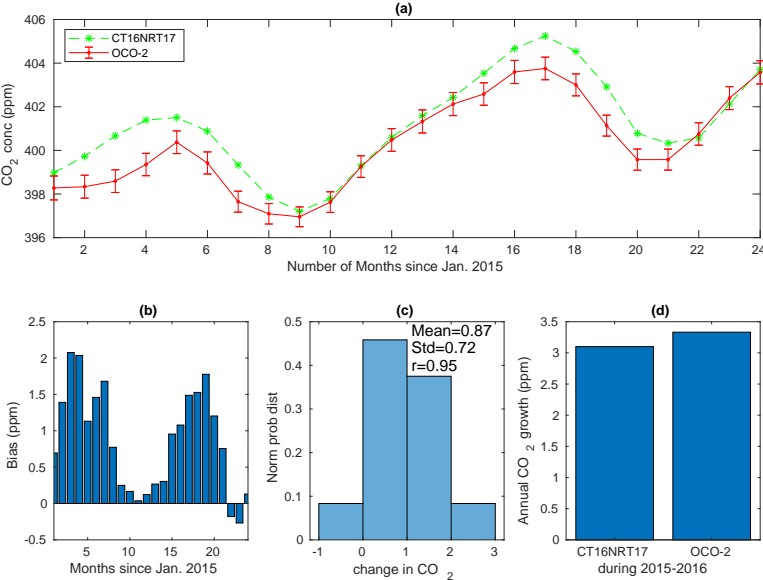

**Figure 14.** The monthly mean time series of CT16NRT17 and OCO-2 from January 2015 to December 2016 averaged over North Africa (a), bias associated to the monthly means (b), the histogram of difference (c) and the annual growth rate obtained by subtracting the mean from the mean of the next year (d). The error bars in (a) shows the OCO-2 a posteriori $XCO_2$ uncertainty.

difference between the two datasets is minimum; On the other hand, a maximum difference of exceeding 1 ppm was observed during MAM which is a burning season in the region (**?**), The observed lower $XCO_2$ values from OCO-2 observations than that of CT16NRT17 simulation will be a consequence of much respiration which exceeded photosynthesis when vegetation uptake is weak following the strong El Niño and dry season over North Africa. Further more, intense burning of during this season

my cause more aerosol loading which will further intensified by of strong El Niño may not sufficiently estimated. Moreover, Fig. 14c displays a monthly mean regional mean bias of 0.87 ppm, correlation of 0.95 and a root mean square deviation of 0.72 ppm between CT16NRT17 and OCO-2 $XCO_2$. This implies that CT16NRT17 is in a good agreement with OCO-2. However, a small discrepancies arose due to a strong anthropogenic emission from Nigeria, Egypt and Algeria together with the establishment of plantation over North Africa, which recently exceeded deforestation, and resulted in net flux of carbon

sink (Canadell et al., 2009). This might have contributed to the observed discrepancy over North Africa.

    Figs. 15a - 16a show monthly mean time series of $XCO_2$ from the model and OCO-2 instrument over Equatorial Africa and Southern Africa which are also in good agreement in terms of pattern. However, the figures show that CT16NRT17 simulations are lower than those of OCO-2 during October, November and December whereas it is opposite during April, May and June over Equatorial Africa and Southern Africa. Figs. 15b and 16b depict a seasonal bias in the monthly time series over Equatorial

Africa and Southern Africa respectively. Positive biases are observed during dry seasons while negative biases are during wet seasons. Moreover, the datasets have monthly averaged regional mean biases of 0.13 and 0.11 ppm, correlation of 0.90 and 0.94, RMSD of 0.84 and 0.73 ppm over Equatorial Africa and Southern Africa respectively. This shows that existence of better





agreement between CT16NRT17 and OCO-2 over these regions in terms of monthly average regional mean values. Figs. 14d-16d show both CT16NRT17 and OCO-2 are in good agreement in estimating the annual growth rate. Patra et al. (2017) found a global mean of more than 3 gigatone of $CO_2$ added to the atmosphere due to the strong El Niño event that occurred during 2015-2016. In agreement with this, both CT16NRT17 and OCO-2 shows an annual growth rate that ranges from 3.10 to 3.42

5    ppm year$^{-1}$ of $XCO_2$ over Africa's land mass (see also Table 8). However, over all regions of Africa's land mass CT16NRT17 shows lower $XCO_2$ annual growth rate than those of OCO-2.

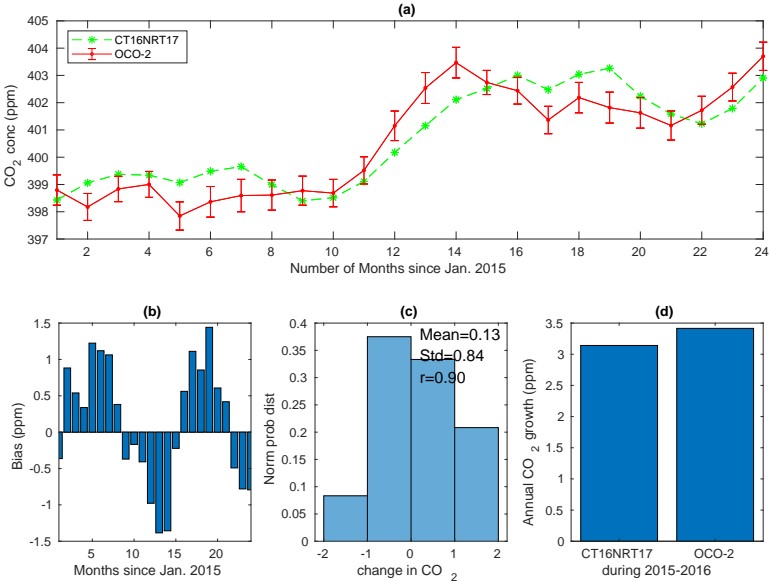

**Figure 15.** The same as in Fig. 14 but over Equatorial Africa.

**Table 8.** Annual growth rate (AGR) of $XCO_2$ over Africa land mass from CT16NRT17 and OCO-2. The results are obtained as the mean annual difference of 2015 and 2016 values

| Region | AGR of CT (ppm year$^{-1}$) | AGR Of OCO-2 (ppm year$^{-1}$) |
|---|---|---|
| North Africa | 3.10 | 3.33 |
| Equatorial Africa | 3.14 | 3.42 |
| Southern Africa | 3.20 | 3.16 |

### 3.7    Comparison of seasonal means of NOAA CT16NRT17 and OCO-2 $XCO_2$

Fig. 17 depicts seasonal means of $XCO_2$ over Africa's land mass from CT16NRT17 (left panels), OCO-2 (middle panels) and their difference (right panels) covering period of January 2015 to December 2016. The white space seen over some regions

10    (e.g., Mali during JJA) is due to insufficient coincident satellite data according to the selection criteria during these seasons.



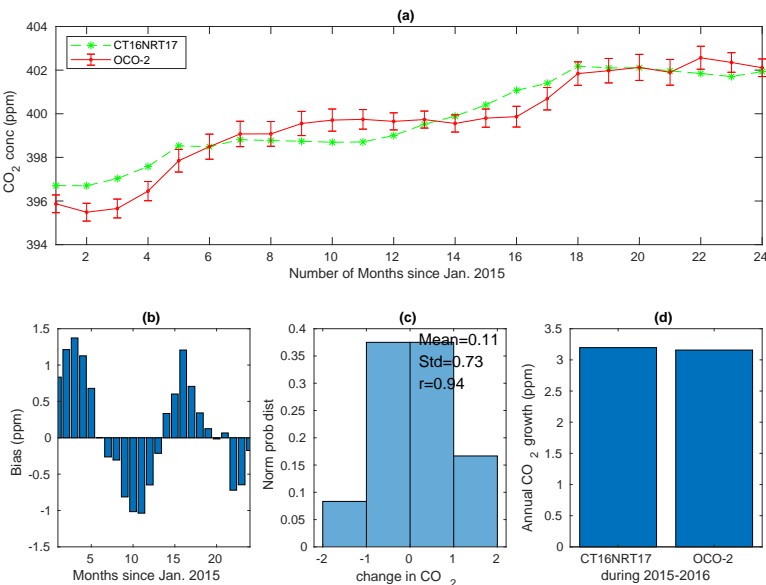

**Figure 16.** The same as in Fig. 14 but over Southern Africa.

$XCO_2$ increases from winter to spring and then decreases from spring peak to summer minimum over the whole continent. The decrease from spring maximum to summer continued into autumn over northern half of Africa in contrast to southern half of Africa which exhibits an increase in $XCO_2$. The decrease from spring to autumn (northward of equator) and until summer (southward of equator) is likely to be a consequence of the land vegetation awakening from dormancy of winter and

partly spring. Conversely, the decomposition of died and decayed vegetation which began in autumn and continued throughout winter adds extra $CO_2$ leading to a maximum concentration during spring (Idso et al., 1999). In agreement with this, both CT16NRT17 and OCO-2 show maximum $XCO_2$ during MAM over North Africa and during SON over Southern Africa. Conversely, minimum concentrations are observed during SON over North Africa and during DJF over South Africa.

Fig. 17 (right panels) shows the seasonal mean difference of CT16NRT17 and OCO-2. A higher mean difference greater

than 1 ppm is observed over North Africa during DJF and MAM when the vegetation cover over the region decreases and also an intensive fire. This indicates that $XCO_2$ values from CT16NRT17 are higher than that of OCO-2 when vegetation uptake is weak and more fire. On the other hand, higher negative mean difference of less than -2 ppm are observed over Equatorial Africa during DJF during SON over Southern Africa. This difference between the CT and OCO-2 arises likely during forest fire that naturally occurs following their respective dry season. Consistent with report by Liang et al. (2017), low seasonal

variability is observed between CT16NRT17 and OCO-2 in the range from -4 to 4 ppm with greater amplitude over North and Equatorial Africa than over Southern Africa (see Fig. 17 (right panels)). During dry seasons OCO-2 over estimates values over the Northern Africa but it underestimates for the Southern Africa.

Fig. 18 shows the histogram of seasonal mean difference of CT16NRT17 and OCO-2. The smaller standard deviation of 1.49 and 1.07 are observed during JJA and SON. On the other hand, higher standard deviation of 1.69 and 1.75 ppm are observed





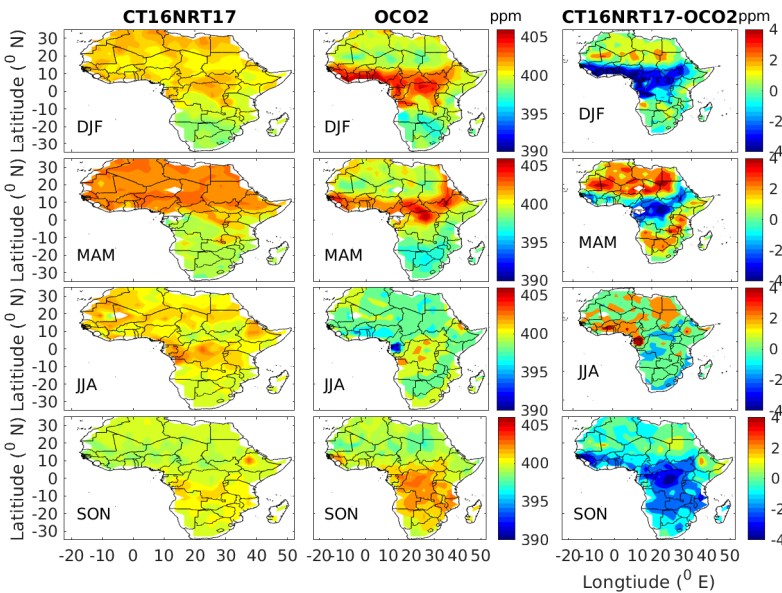

**Figure 17.** Seasonal mean of $CO_2$ for NOAA CT16NRT17 (left panels) and OCO-2 (middle panels) and their difference (right panels).

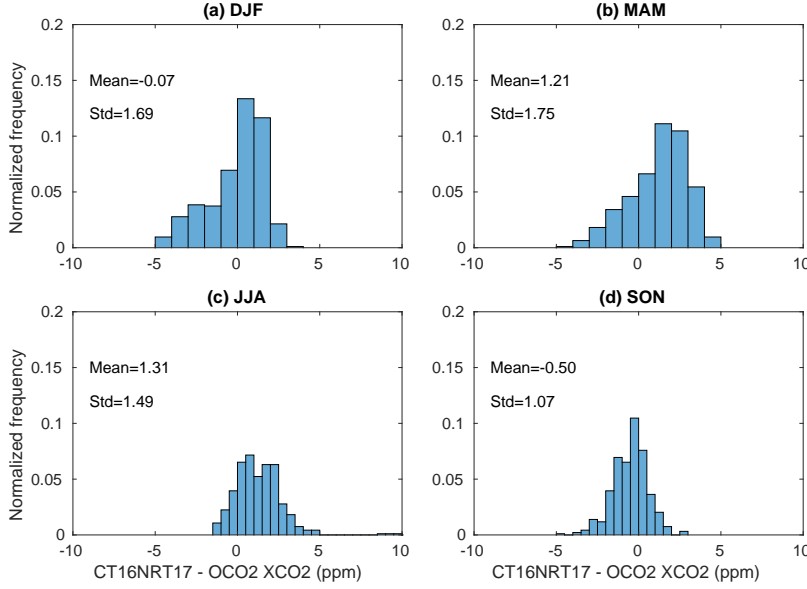

**Figure 18.** Histogram of difference for the seasonal $CO_2$ climatology for DJF (a), MAM(b), JJA (c) and SON (d) seasons.

during DJF and MAM respectively. The results indicate that CT16NRT17 and OCO-2 show a better consistency during wet





seasons and this consistency decreases as the vegetation cover decreases over most regions of Africa land mass during dry seasons.

## 3.8 Comparison of OCO-2 and CT16NRT17 with flask observations

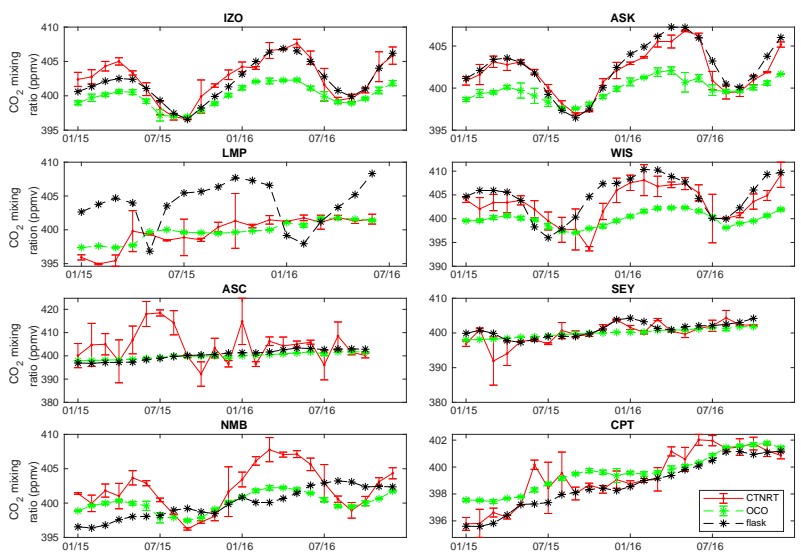

**Figure 19.** $CO_2$ from CT16NRT17, OCO-2 and flask observations.

Monthly CT16NRT17 $XCO_2$ has a better sensitivity over IZO and ASK both in terms of temporal pattern (phase) and
amplitude than OCO-2 (see Fig. 19) where observations from OCO-2 mostly underestimates $XCO_2$ at the two flask sites. Over LMP and WIS, both CT16NRT17 and OCO-2 have moderate sensitivity in capturing the seasonal cycle. On the other hand, OCO-2 has a better sensitivity over ASC and SEY. In addition, $XCO_2$ from both CT16NRT17 and OCO-2 is found to have poor correlations with flask observations over NMB and CPT. However, OCO-2 has closer sensitivity in capturing amplitudes than CT16NRT where CT16NRT17 overestimates $XCO_2$ at these flask sites. In general, CT has a better performance over sites
located at high altitude (IZO, ASK) where satellite observations underestimates $XCO_2$. Conversely, satellite observations have better performance over low altitude island sites (ASC and SEY) as revealed by better agreement with flask $XCO_2$ observations.

## 4  Conclusions

In this study, the tow GOSAT and OCO-2 $XCO_2$ observations values are compared with NOAA CT $XCO_2$ and available
ground based flask observations over Africa land mass. Comparison between GOSAT and CT2016 were done using a five years of datasets covering the period from May 2009 to April 2014. This comparison is important to test the performance of GOSAT in capturing CT and indicating where large discrepancy occurred. Comparison of OCO-2 with CT16NRT17 and



**Table 9.** Summary of statistical relation CT16NRT17 and OCO-2 observation with respect to flask observations. The statistical analysis were made using monthly averaged covering the period from May 2009 to April 2014).

| code | CT R | OCO2 R | CT Bias (ppm) | OCO2 Bias (ppm) | CT RMSD (ppm) | OCO2 RMSD (ppm) | number of data |
|------|------|--------|---------------|-----------------|---------------|-----------------|----------------|
| ASC | -0.14 | 0.97 | 3.93 | -0.48 | 7.63 | 1.10 | 22 |
| ASK | 0.97 | 0.93 | -0.47 | -2.60 | 0.80 | 1.88 | 24 |
| CPT | 0.91 | 0.98 | 0.62 | 0.90 | 0.80 | 0.53 | 24 |
| NMB | 0.28 | 0.42 | 2.14 | 0.09 | 3.27 | 2.02 | 24 |
| IZO | 0.93 | 0.97 | 0.46 | -2.16 | 1.10 | 1.33 | 24 |
| LMP | 0.02 | -0.09 | -4.20 | -4.08 | 3.82 | 3.61 | 18 |
| SEY | 0.68 | 0.71 | -0.98 | -0.98 | 2.23 | 1.47 | 22 |
| WIS | 0.73 | 0.68 | -1.64 | -4.84 | 2.90 | 3.25 | 24 |

eight flask observations was also done using two years data during the strong El Niño event from January 2015 to December 2016. This provides opportunity to assess the performance of OCO-2 Observation during strong El Niño events. Comparison of Carbon Tracker with the two satellites reveals biases of -0.28 and 0.34 ppm, correlations of 0.83 and 0.60 and root mean square deviations of 2.30 and 2.57 ppm with respect to GOSAT and OCO-2 respectively.

The monthly average time series of CT2016 over North Africa, Equatorial Africa and Southern Africa are separately compared with $XCO_2$ from the two satellites. CT2016 agrees well with measurements from the two instruments in terms of pattern and amplitude. However, this agreement deteriorates over Equatorial and Southern Africa in terms of amplitude. It is also found that there is a seasonal dependent bias between them which is negative during dry seasons while it is positive during wet seasons. This indicates results of CT2016 are mostly lower than the GOSAT observation during dry seasons. High spatial mean of

seasonal mean RMSD of 1.91 during DJF and 1.75 ppm during MAM and low RMSD of 1.00 and 1.07 ppm during SON in the model $XCO_2$ with respect to GOSAT and OCO-2 are observed respectively thereby indicating better agreement between CT and the satellites during autumn. CT2016 has the ability to capture monthly time series and seasonal cycles. However, $XCO_2$ from CT2016 is lower than GOSAT observations over North Africa during all seasons whereas $XCO_2$ from CT2016 is higher than that of GOSAT over Equatorial and Southern Africa with the exceptions of DJF over Equatorial Africa and SON over

Southern Africa. In addition, CT2016 simulates lower $XCO_2$ than the observations over some regions (e.g., Congo, South Sudan and southwestern Ethiopia) and during summer season over the whole continent following large vegetation uptake. In contrast, $XCO_2$ from CT16NRT17 is higher than that of OCO-2 over North Africa whereas it is lower than that of OCO-2 during DJF and SON over Equatorial and Southern Africa respectively. Comparison of satellite and CT with ground-based flask observation shows CT has a better performance over sites located at high altitude (IZO, ASK) as determined from good agree-

ment with flask $XCO_2$ observations where satellite observations underestimates $XCO_2$. Conversely, satellite observations have better performance over low altitude sites (ASC and SEY).

    In general, $XCO_2$ from NOAA CT shows a very small bias with respect to GOSAT and OCO-2 observation over Africa's land mass. Moreover, there is a good agreement between CT simulation and observations in terms spatial distribution, monthly





average time series and seasonal climatology. However, there are some discrepancies between the model and the two $XCO_2$ datasets from GOSAT and OCO-2 implying that the accuracy of the model data needs further improvements for the rain forest regions (e.g., Congo) through assimilation of in-situ observations and tuning of the model through process studies.

*Acknowledgements.* The authors acknowledge NOAA Earth System Research Laboratory and NASA GOSAT for the data products. The first

5   author also acknowledges Addis Ababa University, Addis Ababa Science and Technology University, Botswana International University of Science and Technology for their support through fellowship and access to the research facilities.



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
