# Peer review of "On the performance of satellite-based observations of $XCO_2$ in capturing the NOAA Carbon Tracker model and ground-based flask observations over Africa land mass"

_Atmospheric Measurement Techniques, 2019_

## Referee Comment (RC1) · Anonymous Referee #1 · 29 Jan 2020

General comments This is a very timely and very useful study of a much neglected problem – I strongly recommend publication. Scientific observation of CO2 over Africa is extremely limited. Satellites watch the continent, but much of tropical Africa is under heavy cloud in the crucially important high-growth periods in the rainy season. On the ground in situ observation is minimal, and the few sites that are measured are mainly located on remote islands or around the continental periphery. Mengistu and Tsidu tackle this problem by examining the sensitivity and trustworthiness of GOSAT and

[Figure]

OCO satellite measurements, tested against both the NOAA Carbon Tracker model and directly in comparison with the few available flask data sets.

The paper is well thought out, detailed, and careful. There are some problems with the language but these are minor and there is full clarity of meaning.

I strongly recommend publication after minor revision.

Specific Comments Page 1. Line 20 – all over Southern Africa? Does this mean south of the equator? Or south of the Zambesi? Page 2 L4 – space after networks. Page 2 L14 – maybe give more mention to the TCCON station on Ascension Island. Contact D. Feist. https://data.caltech.edu/records/210 I note that ASC is mentioned in table 1. Page 2 L35 onto P3 last sentence doesn't really mean anything. Also note that total column over many places includes very different air masses. For example over Ascension the air under the Trade wind Inversion is from the Southern Ocean and further, while the air above it is from the Congo, and ultimately further away. Page 3 L 1 – say where Kuwalik found this, geographically. Page 3 L13 – African aerosol loading is very seasonal – very bad in biomass burning seasons. Page 3 L 30 – TM5 transport modelling – good. Explain in more detail. Page 4 L23 – maybe explain in more detail about the systematic error Page 4 L25 – I think this means world's second, not 'second world' (i.e. Russia & China). Page 5 Table 1 – Maybe mention the TCCON instrument Leicester have set up at Jinja Uganda (though it will be too late for this paper). Page 7 L8 – southern part of Congo (does this mean Congo Brazzavile???The southern Brazzaville Congo is similar to Kinshasa so I'm puzzled by that comment.) and then the text mentions Southern DRC….note the southern DRC is savanna, not forest, and has intense biomass burning in winter. Page 7 L10 – I am very puzzled by the comment on "weak anthropogenic emissions" from South Africa, which has bigger CO2 emissions than either the UK or France. South Africa has some of the world's biggest CO2 point sources including the enormous SASOL synthetic oil-form-coal plant and many >4GW coal-fired power stations. The ITCZ is critical of course, in two ways – it marks the effective boundary between the two meteorlogical hemispheres, and it

also controls the vegetation uptake, as the plants grow under it, while the fires occur when it is in the opposite end of its range. Page 7 L18 – year-round rainfall only near the coast in West Africa. Inland northern Nigeria is highly seasonal. The forest is only at the southern equatorial frings of this band of countries. Page 7 L29 – note NOAA calibrated measurements are ppm, NOT ppmV. Best to stick to ppm, even through there is only a tiny difference between ppm and ppmv. Page 8 L10 – annual mean position of the ITCZ – this is the meteorological hemisphere boundary. Might be worth expanding this remark. Page 8 L17 – model weakness? Or terrible satellite visibility when the ITCZ is present and clouds are extremely thick and widely present. Page 9 L5 – "satellite own" ?? Typo?? Page 10 L2 – Africa is one of the largest – rewrite as terrible English! I think this means it has more land on both sides of the equator than South America, but I'm not sure! Page 10 L4-13 – maybe move this entire paragraph to a place much earlier in the manuscript, to explain the focus on Africa? Page 12 L15 – "simulation respond" - ??? does this mean response?? Page 13 L14 – sahara – it's a desert! I have flown over it many times. Not a weak source/sink – the vegetation is a nearly zero source/sink but there are very large flaring operations in the Algerian and Libyan oil and gas fields. Those must be big emitters. Page 14 L13 – these are the winter & summer months for the Northern Hemisphere. Opposite in SH. Page 14 L18 – winter (DJF) in Southern Africa???!!!! – last time I heard it was high summer!!! Winter in the Southern Hemisphere is JJA. More to the point, the key factor for vegetation is the distinction between the rainy season (ITCZ present - growth) and the dry season (No ITCZ – fires). Page 16 L2 and L3 – maybe discuss this CT/GOSAT discrepancy in a little more detail? ITCZ cloud blocking observation?? Page 17 L6 CT underestimation – interesting. Page 17 L18 – note Northern Africa includes two very different biomes. North Africa (Morocco, Algerian coast, Tunisia) has a wet Mediterranean winter. The Sahara is desert but has big oil and gasfields, (including supplying Europe with winter gas). Page 19 L3 – note that at the start of an El Nino there is often intense biomass burning. Later, the grass fires are smaller because there is no fuel. Page 23 L2 – Question mark in text??? Which region is the text talking about? – North Africa??

– if so, it is wet in the Algerian mountains in MAM. Fires are in summer. See also Line 4 in same paragraph. Page 23 L5 – "my cause"?? Page 23 L9 – plantation – well, maybe, but I flew over this a while ago and didn't see much! Note that Nigeria is very very different form Egypt, and both are very different from Algeria!!! I think this paragraph needs substantial revision. Page 25 L13 – note that grass fires dominate in the dry savanna, while leaf litter fires are common in the wetter wooded savanna. Page 27 Section 3.8 and Figure 18 – maybe it is worth expanding this section 3.8 very significantly – it has real data!! Also note that these are boundary layer measurements. For example the Trade Wind Inversion (about 1500m in the Atlantic) is really important – ASC is below it, while IZO is well above it, so they sample completely different types of air mass (as noted in the last sentence of the section). General comment on the text Through the text there are many minor language problems. Some sentences are especially challenged grammatically. However, in contrast, many long sections read fluently and clearly. The language infelicities are many but small and not significant – the overall message gets through. The problems could easily be cleared up to make the work easier to read.

Conclusion This is a valuable and very interesting study. The paper should certainly be published, but it needs minor revision.

---

## Referee Comment (RC2) · Anonymous Referee #2 · 7 Feb 2020

General comments:

The manuscript entitled, "On the performance of satellite-based observations of CO2 in capturing the NOAA Carbon Tracker model and ground-based flask observations over Africa land mass" presents a scientifically interesting comparison of Carbon Tracker, GOSAT, OCO-2, and flask CO2 measurements. Despite Africa lacking ground-truth instruments such as TCCON, studies such as this one are useful for pointing out differences in the models and satellite observations. In general, there is one major methodological issue and many clarifications and technical fixes needed, but I recommend publication once they are resolved.

General comments:

- GOSAT and OCO-2's primary product is the column-averaged dry-air mole fraction of CO2 (XCO2), not a vertical profile of CO2. There are typically less than 2 degrees of freedom for vertical CO2 for any given retrieval. Thus, the entire comparison to flasks should come with a disclaimer that the NASA L2 retrievals for GOSAT and OCO-2 are not designed to be used in this way. The comparison is still interesting, but I am unsure about the scientific value.

- The authors often list characteristics of a certain region (e.g. high anthropogenic emissions, low vegetation levels) and then attribute the difference between CT and GOSAT/OCO-2 to these characteristics. The data is indicating correlation, not causation. Additional research (e.g. a detailed modelling study) would need to be done to provide evidence that the XCO2 difference is *caused* by such characteristics. I note several instances of this below where it would be wise to soften the language.

- For all the maps, I would strongly suggest not to use the default rainbow colormap for XCO2. Depending on the coding language you use, there are a number of much better colormaps available. For ordered information, such as XCO2, you should use a perceptually uniform colormap (such as viridis in Python). For diverging data, such as CT2016 – GOSAT, you should use a diverging colormap (such as RdBu in Python) and center the colorbar at 0. In many of your figures, you use a rainbow colormap with unequal positive and negative limits, which makes it incredibly difficult to determine where on the map the bias is above or below zero.

https://matplotlib.org/tutorials/colors/colormaps.html

- When discussing the distance between a given GOSAT/OCO-2 measurement and CT, could you please elaborate on what exactly this means? Each GOSAT/OCO-2 measurement should fall within a CT grid cell, so dx seems meaningless to me.

- The mean bias for the entirety of Africa is mentioned numerous times, including in the abstract. However, your analysis shows that there are large regional patterns. Thus, there is little scientific value in, for example, stating that GOSAT XCO2 is 0.28 ppm higher than CT. Additionally, no uncertainties are given for any statistics in this paper. This should be resolved

before publication. For example, 0.28 +/- 1.5 ppm is much less meaningful than 0.28 +/- 0.2 ppm.

- For OCO-2, are you using land nadir data, land glint data, or both? For GOSAT, you are presumably including the medium gain data, but please state so.

Specific comments:

- P2 L30: Citation for this? The land surface characteristics could affect retrievals, but I'm unaware of the impact of anthropogenic sources on satellite XCO2 biases.

- P3 L9: This makes it sound as if models are intrinsically more accurate than the satellite measurements. If this were true, why would we even need satellite measurements? In general, however, the paper does a good job at saying the models and obs. "agree" or "disagree" rather than one is "wrong" or "right."

- P4 L10: SCIAMACY measured CO2 and CH4 before GOSAT.

- P4 L19: GOSAT ACOS B3.5 is now ~5.5 years out of date. B7.3, which represents a significant update to the retrieval, has been available for over 3 years now. It is too much to ask of the authors to repeat their analysis with the newer version, but it must be noted that the version used is very outdated. See the official Data Users Guide for details on the latest product:

https://docserver.gesdisc.eosdis.nasa.gov/public/project/OCO/ACOS_v7.3_DataUsersGuide-RevF.pdf

- P4 L26: Please cite some OCO-2 papers in this section (e.g. Crisp et al., 2008,

- P5 L16: If CT is a 3-hourly product, the maximum d(time) would be 1.5 hours.

- P7 L10: Citation needed regarding Southern Africa's characterization.

- P7 L11: How do you know that this is the reason for the bias dipole?

- P7 L19: How would low number statistics result in a high bias? It's certainly possible, but no explanation or mechanism is provided.

- P7 L19: Citation needed regarding rainfall.

- P8 L1: These plots are very difficult to interpret because of the large number of data points. I would strongly suggest to instead plot heatmaps of the XCO2 difference vs. the spatial difference. And, as noted above, it is not clear what the distance metric actually represents.

- P9 L5: The higher GOSAT/OCO-2 uncertainty in these regions is likely driven by low signal to noise in the strong CO2 band over dark forests.

- P10 L6: Could use a general citation here.

- P12 L15: If the CO2 sink is growing after the rainy season, why would GOSAT not see it?

- P14 L1: Same as above: why would there be a difference? You seem to imply that the difference must be because of local sources and transport, yet this is speculation. I would simply soften the language from "likely" to "possibly."

- P17 L4: The cirrus cloud hypothesis should be removed unless you can show that there are more cirrus clouds over that specific region which could potentially be biasing the satellite results.

- P17 L11: By what mechanism would a cold bias impact the CT XCO2? Would suggest removing unless you can provide a reasonable hypothesis.

- P17 L18: How would low vegetation levels and local sources result in a low correlation between the two products? Would suggest removing unless you can provide a reasonable hypothesis.

- P19 L17: Good. Here, a correlation is discussed (higher OCO-2 where there's more vegetation) without asserting causation. Another hypothesis could be cloud contamination in the satellite retrievals.

- P23 L9: What plantation is this referring to? Please elaborate or remove this statement.

- P25 L11: What intensive fire is this referring to? Please elaborate or remove this statement.

- P29 L2: This is a disappointingly brief discussion on reasons why the model could have issues. This paper should emphasize that neither models nor satellites are perfect, and that all that can be done in a poorly constrained place such as Africa is a comparison and discussion of potential reasons for the differences. For example, clouds, aerosols, and dark surfaces can result in biased XCO2 from satellites, while poor parameterizations and insufficient input data can hinder models.

- P29 L4: Should thank both the appropriate Japanese agencies for GOSAT and NASA JPL for the GOSAT ACOS and OCO-2 retrievals.

Technical comments:

There are numerous spelling and grammar issues that should not be the responsibility of a reviewer to fix. I would suggest that the authors spend some time resolving these issues.

Overall: $XCO_2$ is never defined.

- P3 L25: "combines observed in situ carbon dioxide"
- P7 L15: Likely a typo. GOSAT in comparison to GOSAT.
- P10 L2: Oddly worded. Just say Africa has significant land mass in both hemispheres.
- P27 L17: Oddly worded. Perhaps, "is important to identify differences between GOSAT and CT."

Figure comments:

- As stated above, please use appropriate colormaps and colorbar ranges for diverging data.
- For time series, please use years and months instead of "months since."

---

## Author Comment (AC1) · 4 Apr 2020

Authors Response to Anonymous Referee #1 comments and suggestions on manuscript entitled " On the performance of satellite-based observations of CO2 in capturing the NOAA Carbon Tracker model and ground-based flask observations over Africa land mass " by Anteneh Getachew Mengistu and Gizaw Mengistu Tsidu

We thank both Anonymous Reviewers, for their time and constructive comments on our

manuscript. These comments are very helpful to improve the quality of the manuscript and therefore we have carefully used them to revise the manuscript.

Comments: This is a very timely and very useful study of a much neglected problem – I strongly recommend publication. Scientific observation of CO2 over Africa is extremely limited. Satellites watch the continent, but much of tropical Africa is under heavy cloud in the crucially important high-growth periods in the rainy season. On the ground in situ observation is minimal, and the few sites that are measured are mainly located on remote islands or around the continental periphery. Mengistu and Tsidu tackle this problem by examining the sensitivity and trustworthiness of GOSAT and Printer-friendly version Discussion paper OCO satellite measurements, tested against both the NOAA Carbon Tracker model and directly in comparison with the few available flask data sets.

Response: We thank the Anonymous Referee for his acknowledgment that our study is timely and useful. We hope that this study will bring attention to the regional study and strengthening the carbon network in Africa.

Comments: The paper is well thought out, detailed, and careful. There are some problems with the language but these are minor and there is full clarity of meaning. I strongly recommend publication after minor revision. Response: We have made some efforts to improve the language used in the manuscript and increases its readability. Specific Comments: Page 1. Line 20 – all over Southern Africa? Does this mean south of the equator? Or south of the Zambesi?

Response: In the main text we define the regions as: Northern Africa (100 - 350 N), Equatorial Africa (100 S – 100 N) and Southern Africa (350 - 100 S). However, we did not mention it in the abstract. Now we update the text in the abstract to describe the region boundary. Rightly so, southern Africa refers to a region south of Zambesi. Change is made on page 1 of line 21.

Specific Comments: Page 2 L4 – space after networks.

Response: Change is implemented.

Specific Comments: Page 2 L14 – maybe give more mention to the TCCON station on Ascension Island. Contact D. Feist. https://data.caltech.edu/records/210 I note that ASC is mentioned in table 1. Response: The TCCON station on Ascension Island is mentioned as an example of the TCCON stations. Change is made on page 2 of lines 18-21.

Specific Comments: Page 2 L35 onto P3 last sentence doesn't really mean anything. Also note that total column over many places includes very different air masses. For example over Ascension the air under the Trade wind Inversion is from the Southern Ocean and further, while the air above it is from the Congo, and ultimately further away.

Response: The statement on Page 2 L35 gives information that validation studies are important and had been also conducted by other researchers. It shows further that the results they have obtained at a global and regional scale elsewhere which will give the expected accuracies from our study. And Page 3 of the last line provides information about the TM5 model resolution on the global and North America which can give a clue for readers about the limitation of CT on a global scale as it has a sparse resolution due to the transport model used. These statements are now on page 3 of line 6 and page 4 of line 16.

Specific Comments: Page 3 L 1 – say where Kuwalik found this, geographically.

Response: The comparison study in the work of Kuwalik et al. was done using 17 different TCCON sites across the globe. We updated the text as "relative to 17 TCCON sites across the globe...". This change is made on page 3 of line 8.

Specific Comments: Page 3 L13 – African aerosol loading is very seasonal – very bad in biomass burning seasons. Response: Thanks for reminding us of the importance of seasonal aerosol loading's beside the geographical variation. We update the text as: "In addition, seasonal variation of biomass burning in Africa...." change is made on

page 3 line 19.

Specific Comments: Page 3 L 30 – TM5 transport modelling – good. Explain in more detail.

Response: accepted and updated. See page 4 of line 17 "The model can be used in a wide range of applications, which includes aerosol modeling...."

Specific Comments: Page 4 L23 – maybe explain in more detail about the systematic error.

Response: accepted and updated as: "Chevallier (2015) shows systematic error in the African savanna associated with underestimating the intensity of fire during March at the end of the savanna burning season". This change is made on page 5 of line 10.

Specific Comments: Page 4 L25 – I think this means world's second, not 'second world' (i.e. Russia & China).

Response: thank you for noting this. Now it is corrected on page 5 of line 14.

Specific Comments: Page 5 Table 1 – Maybe mention the TCCON instrument Leicester have set up at Jinja Uganda (though it will be too late for this paper).

Response: Thank you for suggesting the newly established TCCON site in Uganda. This site will be a promising data source for future studies. We indicated the presence of this site in the introduction section of the revised manuscript as potential site that can provide data to bridge existing data gaps in the future.

Specific Comments: Page 7 L8 – southern part of Congo (does this mean Congo Brazzavile??? The southern Brazzaville Congo is similar to Kinshasa so I'm puzzled by that comment.) and then the text mentions Southern DRC....note the southern DRC is savanna, not forest, and has intense biomass burning in winter.

Response: It was the Congo Brazzaviel to increase clarity we updated the text as: "some part of Equatorial Guinea and the Republic of Congo for CT (Fig. 1a) and part

of Democratic Republic of Congo for GOSAT (Fig. 1b)" . This change is made on page 7 of Line 22.

Specific Comments: Page 7 L10 – I am very puzzled by the comment on "weak anthropogenic emissions" from South Africa, which has bigger CO2 emissions than either the UK or France. South Africa has some of the world's biggest CO2 point sources including the enormous SASOL synthetic oil-form-coal plant and many >4GW coal-fired power stations. The ITCZ is critical of course, in two ways – it marks the effective boundary between the two meteorological hemispheres, and it also controls the vegetation uptake, as the plants grow under it, while the fires occur when it is in the opposite end of its range.

Response: Here we compare the Northern and Southern Africa (not South Africa). We agree that South Africa is the biggest fuel source and CO2 emissions from fossil fuels and cement production on continental wise. However, the aggregated emission from countries in Northern Africa like Egypt, Algeria, Nigeria, Libya and Morocco with a large contribution of CO2 emission exceeded South Africa. As a result, the aggregate emission of CO2 from the Northern part of Africa is more than that of Southern Africa.

Specific Comments: Page 7 L18 – year-round rainfall only near the coast in West Africa. Inland northern Nigeria is highly seasonal. The forest is only at the southern equatorial frings of this band of countries. Response: Thank you we made them specific to the southern part of these countries. "southern Guinea, southern Ghana, southern Nigeria, southeast of Central Africa, . . ." change is made on page 8 of line 5.

Specific Comments: Page 7 L29 – note NOAA calibrated measurements are ppm, NOT ppmV. Best to stick to ppm, even though there is only a tiny difference between ppm and ppmv.

Response: Thank you for noting this. It is a type error as noted in the x label of Fig. 2a it is in units of ppm not ppmv. It is now updated on page 9 of line 2.

Specific Comments: Page 8 L10 – annual mean position of the ITCZ – this is the meteorological hemisphere boundary. Might be worth expanding this remark.

Response: accepted and updated as "Position of ITCZ is the main climatic mechanisms controlling rainfall in Africa. Systematic errors due to ITCZ and the East African Monsoon need to be addressed well in satellite retrievals and modeling works." on page 9 lines 3-6.

Specific Comments: Page 8 L17 – model weakness? Or terrible satellite visibility when the ITCZ is present and clouds are extremely thick and widely present.

Response: Thank you for the suggestion, we updated it on page 9 of line 9.

Specific Comments: Page 9 L5 – "satellite own" ?? Typo?? Response: Revised as: "Satellite retrieval uncertainty" on page 10 of line 14.

Specific Comments: Page 10 L2 – Africa is one of the largest – rewrite as terrible English! I think this means it has more land on both sides of the equator than South America, but I'm not sure! Response: Thank you, this statement has been removed in the revised manuscript.

Specific Comments: Page 10 L4-13 – maybe move this entire paragraph to a place much earlier in the manuscript, to explain the focus on Africa?

Response: Thank you. We have now moved this paragraph to introduction as suggested on page 3 of line 21-33.

Specific Comments: Page 12 L15 – "simulation respond" - ??? does this mean response??

Response: It now reads "simulation is more sensitive to ... " on page 12 line 8.

Specific Comments: Page 13 L14 – sahara – it's a desert! I have flown over it many times. Not a weak source/sink – the vegetation is a nearly zero source/sink but there are very large flaring operations in the Algerian and Libyan oil and gas fields.

[Figure]

Those must be big emitters.

Response: It appears that the text did not convey the required message as our intention is to emphasis local emission. Therefore, we rewrote it as "This is mainly because Northern Africa is dominated by the Sahara desert, which is a vegetation-free area, and the systematic bias due to the local atmosphere biosphere interaction is minimum. However, the spatial mean of monthly mean bias is slightly higher (-0.36 ppm) over North Africa than over Equatorial Africa (-0.17 ppm) and Southern Africa (0.01 ppm). This is likely due to the presence of strong local emissions from Egypt, Algeria, and Libya as well due to long-range transport from the Northern Hemisphere..." on page 14 of lines 7-13.

Specific Comments: Page14 L13 – these are the winter & summer months for the Northern Hemisphere. Opposite in SH.

Response: We agree that it is good to mention that they are for the Northern Hemisphere and the opposite is for the southern hemisphere. Change is made on page 15 of line 10.

Specific Comments: Page14L18– winter (DJF) in Southern Africa???!!!! – Last time I heard it was high summer!!! Winter in the Southern Hemisphere is JJA. More to the point, the key factor for vegetation is the distinction between the rainy season (ITCZ present - growth) and the dry season (No ITCZ – fires).

Response: Thank you for highlighting our silly mistake. It is corrected on page 15 of line 10.

Specific Comments: Page 16 L2 and L3 – maybe discuss this CT/GOSAT discrepancy in a little more detail? ITCZ cloud blocking observation??

Response: We hope that it has been discussed sufficiently on the next paragraph on page 16 line 8 - 18.

Specific Comments: Page17 L6 CT under estimation – interesting. Page 17 L18 – note

Northern Africa includes two very different biomes. North Africa (Morocco, Algerian coast, Tunisia) has a wet Mediterranean winter. The Sahara is desert but has big oil and gas fields, (including supplying Europe with winter gas).

Response: accepted and changes are made to highlight the differences between these places.

Specific Comments: Page 19 L3 – note that at the start of an El Nino there is often intense biomass burning. Later, the grass fires are smaller because there is no fuel.

Response: accepted and change is made to reflect this process.

Specific Comments: Page 23 L2 – Question mark in text??? Which region is the text talking about? – North Africa?? – if so, it is wet in the Algerian mountains in MAM. Fires are in summer. See also Line 4 in same paragraph. Response: Thank you. The question mark in the text is due to a missed citation in compiling the Latex. Now we include the reference. We know that regions of Africa have different burning seasons but the reference listed refers to the burning seasons of Africa in the context of the general areas in the north and south of the equator. Change has been made on page 23 line 16 and page 24 line 1.

Specific Comments: Page 23 L5 – "my cause"??

Response: Corrected as "may cause" on page 24 line 4.

Specific Comments: Page 23 L9 – plantation – well, maybe, but I flew over this a while ago and didn't see much! Note that Nigeria is very different form Egypt, and both are very different from Algeria!!! I think this paragraph needs substantial revision.

Response: Thank you for sharing your observation of the region. We updated the statement on page 24 of line 7.

Specific Comments: Page 25 L13 – note that grass fires dominate in the dry savanna, while leaf litter fires are common in the wetter wooded savanna.

Response: Thank you for the suggestion. Our observation shows the discrepancy during the dry season and so it is most likely due to grass fries from the dry savanna. Now the text is updated in this sense on page 26 from line 9-10.

Specific Comments: Page 27 Section 3.8 and Figure 18 – maybe it is worth expanding this section 3.8 very significantly–it has real data!! Also note that these are boundary layer measurements. For example the Trade Wind Inversion (about 1500m in the Atlantic) is really important – ASC is below it, while IZO is well above it, so they sample completely different types of air mass (as noted in the last sentence of the section).

Response: We have tried out to further expand the discussion on this section 3.8. Page 27.

Specific Comments: General comment on the text Through the text there are many minor language problems. Some sentences are especially challenged grammatically. However, in contrast, many long sections read fluently and clearly. The language infelicities are many but small and not significant – the overall message gets through. The problems could easily be cleared up to make the work easier to read.

Response: Efforts are made to improve the language in the revised manuscript.

Specific Comments: AMTD Interactive comment Conclusion. This is a valuable and very interesting study. The paper should certainly be published, but it needs minor revision.

Response: Thank you for your recommendation of the work for publication in AMT.

Anteneh Getachew Mengistu and Gizaw Mengistu Tsidu

Please also note the supplement to this comment:
https://www.atmos-meas-tech-discuss.net/amt-2019-390/amt-2019-390-AC1-supplement.pdf

[Figure]

**Supplement:**

[revised manuscript text omitted]

---

## Author Comment (AC2) · 4 Apr 2020

Authors Response to Anonymous Referee #2 comments and suggestions on manuscript entitled " On the performance of satellite-based observations of CO2 in capturing the NOAA Carbon Tracker model and ground-based flask observations over Africa land mass " by Anteneh Getachew Mengistu and Gizaw Mengistu Tsidu

General comments: The manuscript entitled, "On the performance of satellite-based

observations of CO2 in capturing the NOAA Carbon Tracker model and ground-based flask observations over Africa land mass" presents a scientifically interesting comparison of Carbon Tracker, GOSAT, OCO-2, and flask CO2 measurements. Despite Africa lacking ground-truth instruments such as TCCON, studies such as this one are useful for pointing out differences in the models and satellite observations. Response: We thank the anonymous referee for supporting the importance of the study.

General comments: In general, there is one major methodological issue and many clarifications and technical fixes needed, but I recommend publication once they are resolved. Response: We have carefully addressed the comments and suggestions raised by the referee and improved the quality of the manuscript.

General comments: - GOSAT and OCO-2's primary product is the column-averaged dry-air mole fraction of CO2 (XCO2), not a vertical profile of CO2. There are typically less than 2 degrees of freedom for vertical CO2 for any given retrieval. Thus, the entire comparison to flasks should come with a disclaimer that the NASA L2 retrievals for GOSAT and OCO-2 are not designed to be used in this way. The comparison is still interesting, but I am unsure about the scientific value.

Response: Here, we try to include information on the CO2 profile and estimate near-surface values of CO2 mixing ratio to compare the Level 2 data sets of GOSAT and OCO-2 with the flasks values. The XCO2 from the GOSAT and OCO-2 was the column averaged with profile information from top to surface and we have used the lower pressure levels from the satellite retrieval. This kind of comparison of in-situ CO2 measurements and XCO2 retrieved from satellite will provide information on how strong is the influence of the local CO2 flux. The scientific values of comparison of in-situ CO2 measurements with Satellite XCO2 was described in the study of Ye Yuan et.al. 2019 and our study is not for the first time in this sense. General comments: The authors often list characteristics of a certain region (e.g. high anthropogenic emissions, low vegetation levels) and then attribute the difference between CT and GOSAT/OCO-2 to these characteristics. The data is indicating correlation, not causation. Additional

research (e.g. a detailed modelling study) would need to be done to provide evidence that the XCO2 difference is *caused* by such characteristics. I note several instances of this below where it would be wise to soften the language.

Response: We agree with the referee's comment that additional studies are needed to identify and quantify the causes of the discrepancies observed. It is not the scope of this study to quantify all sources of the discrepancy. We have merely indicate some possible source of discrepancy based on physical connection, not just on correlation. Identification of causality chain is complex and may need modeling works in some cases and it is not our intension to do so.

General comments: For all the maps, I would strongly suggest not to use the default rainbow colormap for XCO2. Depending on the coding language you use, there are a number of much better colormaps available. For ordered information, such as XCO2, you should use a perceptually uniform colormap (such as viridis in Python). For diverging data, such as CT2016 – GOSAT, you should use a diverging colormap (such as RdBu in Python) and center the colorbar at 0. In many of your figures, you use a rainbow colormap with unequal positive and negative limits, which makes it incredibly difficult to determine where on the map the bias is above or below zero. https://matplotlib.org/tutorials/colors/colormaps.html Response: We understand the concern of the reviewer. It is always a difficult task in Matlab. We accept the anonymous referee suggestion to enhance the quality of the figures.

General comments: When discussing the distance between a given GOSAT/OCO-2 measurement and CT, could you please elaborate on what exactly this means? Each GOSAT/OCO-2 measurement should fall within a CT grid cell, so dx seems meaningless to me. Response: we averaged satellite values in a 3 X 3 degree window centering the grid cell of CT as described on page 6 line 5. Hence, we use a rectangle the maximum distance of the observation from the satellites can have a value $\sqrt{(ãĂŰ1.5ãĂŮ^2 ãĂŰ+1.5ãĂŮ^2)}$ =2.1 degree which is indicated on the color bar of Fig. 2.

General comments: The mean bias for the entirety of Africa is mentioned numerous times, including in the abstract. However, your analysis shows that there are large regional patterns. Thus, there is little scientific value in, for example, stating that GOSAT XCO2 is 0.28 ppm higher than CT. Additionally, no uncertainties are given for any statistics in this paper. This should be resolved before publication. For example, 0.28 +/- 1.5 ppm is much less meaningful than 0.28 +/- 0.2 ppm. Response: We have indicated the standard deviation of the mean bias in table 1 on page 10. However, We agreed that it was also good to indicate as +/- from the mean bias as suggested. And now we updated in the main text including the abstract.

General comments: For OCO-2, are you using land nadir data, land glint data, or both? For GOSAT, you are presumably including the medium gain data, but please state so. Response: We use both nadir data and land glint data in the analysis as they are both can normally be used for scientific analysis (see Wunch et., al. ). It is explicitly stated on page 5 of line 20 in the revised manuscript.

Specific comments: P2 L30: Citation for this? The land surface characteristics could affect retrievals, but I'm unaware of the impact of anthropogenic sources on satellite XCO2 biases. Response: accepted and citation is added on page 3 of line 2.

Specific comments: P3 L9: This makes it sound as if models are intrinsically more accurate than the satellite measurements. If this were true, why would we even need satellite measurements? In general, however, the paper does a good job at saying the models and obs. "agree" or "disagree" rather than one is "wrong" or "right." Response: The statement on page 3 of lines 7 -11 now on page 3 from lines 13-17 shows the regional uncertainties in GOSAT retrieval varied from one region to others. The GOSAT retrievals did a good job over the US while it has large regional variation over China which suggests the need for consistency check on the satellite retrievals. Our study shows that there are certain limitations and strengths of both models and satellite data.

Specific comments: P4 L10: SCIAMACY measured CO2 and CH4 before GOSAT.

[Figure]

Response: We mentioned GOSAT as the world's first spacecraft dedicated fully to measure the concentrations of carbon dioxide and methane. This statement is re-phrased in this sense on page 4 line 7. SCIAMACY on ENVISAYT is providing good data on CO2 in recent times but it was not CO2 dedicated satellite mission.

Specific comments: P4 L19: GOSAT ACOS B3.5 is now 5.5 years out of date. B7.3, which represents a significant update to the retrieval, has been available for over 3 years now. It is too much to ask of the authors to repeat their analysis with the newer version, but it must be noted that the version used is very outdated. See the official Data Users Guide for details on the latest product: https://docserver.gesdisc.eosdis.nasa.gov/public/project/OCO/ACOS_v7.3_DataUsersGuideRevF.pdf Response: We have specified the data version which can indicate when the datasets were retrieved.

Specific comments: P4 L26: Please cite some OCO-2 papers in this section (e.g. Crisp et al., 2008, Response: accepted and change is made on page 4 of line 15.

Specific comments: P5 L16: If CT is a 3-hourly product, the maximum d(time) would be 1.5 hours. Response: we agree that the maximum d(time ) in CT is 1.5 hour . But instead of 1.5 hrs sampling interval, we used 3 hr to get more coincident measurements.

Specific comments: P7 L10: Citation needed regarding Southern Africa's characterization. Response: accepted and change is effected on page 7 line 25.

Specific comments: P7 L11: How do you know that this is the reason for the bias dipole? Response: The distribution map shows that there is dipole distribution which is higher XCO2 north of the equator than south of the equator. The Southern Africa region is characterized by weak anthropogenic CO2 emission and high CO2 uptake by the vegetation than Northern Africa (see also Ciais et al., 2011).

Specific comments: P7 L19: How would low number statistics result in a high bias?

[Figure]

It's certainly possible, but no explanation or mechanism is provided. Response: That is likely because the satellite retrievals have noise which can be smoothed out when a large number of datasets are averaged.

Specific comments: P7 L19: Citation needed regarding rainfall. Response: accepted and change is made on page 8 line 7.

Specific comments: P8 L1: These plots are very difficult to interpret because of the large number of data points. I would strongly suggest to instead plot heatmaps of the XCO2 difference vs. the spatial difference. And, as noted above, it is not clear what the distance metric actually represents. Response: accepted.

Specific comments: P9 L5: The higher GOSAT/OCO-2 uncertainty in these regions is likely driven by low signal to noise in the strong CO2 band over dark forests. P10 L6: Could use a general citation here. Response: This part is removed and partly considered on the introduction section as recommend by the other referee.

Specific comments: P12 L15: If the CO2 sink is growing after the rainy season, why would GOSAT not see it? Response: This discrepancy is over the African equatorial region which largely covered by dense forests since GOSAT may have large uncertainty over the dark forest region. However, further studies are needed to answer specifically why the discrepancy occurs.

Specific comments: P14 L1: Same as above: why would there be a difference? You seem to imply that the difference must be because of local sources and transport, yet this is speculation. I would simply soften the language from "likely" to "possibly." Response: accepted.

Specific comments: P17 L4: The cirrus cloud hypothesis should be removed unless you can show that there are more cirrus clouds over that specific region which could potentially be biasing the satellite results. Response: accepted and the statement is removed.
Specific comments: P17 L11: By what mechanism would a cold bias impact the CT XCO2? Would suggest removing unless you can provide a reasonable hypothesis. Response: accepted and it is now removed.

Specific comments: P17 L18: How would low vegetation levels and local sources result in a low correlation between the two products? Would suggest removing unless you can provide a reasonable hypothesis. Response: On a vegetation-free area, the XCO2 has weak to no seasonal patterns. Furthermore, the presence of a point CO2 emission source may not be captured by the coarse model simulation.

Specific comments: P19 L17: Good. Here, a correlation is discussed (higher OCO-2 where there's more vegetation) without asserting causation. Another hypothesis could be cloud contamination in the satellite retrievals. P23 L9: What plantation is this referring to? Please elaborate or remove this statement. Response: accepted and the statement was removed.

Specific comments: P25 L11: What intensive fire is this referring to? Please elaborate or remove this statement. Response: The statement is further elaborated on page 26 line 7.

Specific comments: P29 L2: This is a disappointingly brief discussion on reasons why the model could have issues. This paper should emphasize that neither models nor satellites are perfect, and that all that can be done in a poorly constrained place such as Africa is a comparison and discussion of potential reasons for the differences. For example, clouds, aerosols, and dark surfaces can result in biased XCO2 from satellites, while poor parameterizations and insufficient input data can hinder models. Response: Although we are clear on how both observations and model go wrong, we made further statements regarding potential problems in both cases in the manuscript by highlighting reviewer's inputs at various places in the revised manuscript.

Specific comments: P29 L4: Should thank both the appropriate Japanese agencies for GOSAT and NASA JPL for the GOSAT ACOS and OCO-2 retrievals. Technical

comments: There are numerous spelling and grammar issues that should not be the responsibility of a reviewer to fix. I would suggest that the authors spend some time resolving these issues. Response: Changes are made according to the recommendations.

Specific comments: Overall: XCO2 is never defined. Response: accepted and it is defined on page 1 line 4 (abstract) and page 3 line 1. Specific comments: P3 L25: "combines observed in situ carbon dioxide"; P7 L15: Likely a typo. GOSAT in comparison to GOSAT. Response: Changed to "GOSAT ....in comparison to CT" on page 8 line 3.

Specific comments: P10 L2: Oddly worded. Just say Africa has significant land mass in both hemispheres. Response: This paragraph have been moved to introduction and modified on page 3 line 19.

Specific comments: P27 L17: Oddly worded. Perhaps, "is important to identify differences between GOSAT and CT. Response: Accepted and change is made on page 28 line 11.

Specific comments: " Figure comments: - As stated above, please use appropriate colormaps and colorbar ranges for diverging data. - For time series, please use years and months instead of "months since." Response: accepted.

Anteneh Getachew Mengistu and Gizaw Mengistu Tsidu

Please also note the supplement to this comment:
https://www.atmos-meas-tech-discuss.net/amt-2019-390/amt-2019-390-AC2-supplement.pdf